# Influence of Process Parameters on Flatness During Single-Track Laser Cladding

**DOI:** 10.3390/ma17215225

**Published:** 2024-10-26

**Authors:** Guozheng Quan, Haitao Wang, Wenjing Ran, Fanxin Meng

**Affiliations:** Chongqing Key Laboratory of Advanced Mold Intelligent Manufacturing, School of Material Science and Engineering, Chongqing University, Chongqing 400044, China

**Keywords:** laser cladding, flatness, molten pool morphology, parameter window

## Abstract

During the laser cladding process, poor flatness of the cladding track can cause the surface structure to be uneven or corrugated, affecting the geometrical accuracy of the workpiece. Adjusting process parameters is an effective way to achieve high cladding track flatness. This study established a mesoscale model of the laser cladding process for CoCrMoSi powder to simulate the formation of a single cladding track. Subsequently, the formation mechanism of cladding track flatness was revealed by analyzing the flow within the molten pool and the solidification behavior of the molten pool edge. The influences of laser power, scanning speed, and powder feeding rate on flatness were determined through simulations and physical experiments. Finally, a parameter window of flatness was established using simulation and experimental results. The window indicates that high flatness is achieved with a high scanning speed (*v* > 260 mm/min), high laser power (*P* > 2300 W), and low powder feed rate (*P_f_* < 5.5 g/min). The accuracy of the numerical model was verified by comparing the simulated results with the experimental measurements.

## 1. Introduction

Laser cladding technology possesses the advantages of low dilution rate, high density, high bonding strength, and fine grains; consequently, it has been widely used in component manufacturing for aerospace, mold, machinery and shipbuilding [1,2,3,4]. The 40Cr10Si2Mo alloy is a material that is widely used in the manufacture of valves for low-speed engines. However, as the power of low-speed engines increases, the operating conditions of these valves becomes more demanding [5,6]. The current method of controlling grain size to improve valve performance through the electroforming–forging process is no longer sufficient to meet these stringent requirements [7]. Therefore, the use of laser cladding of CoCrMoSi iron-based high-temperature alloys on valve surfaces can effectively improve valve performance. During the laser cladding process, metal powder is incrementally deposited on the metal substrate track by track, layer by layer, under a laser heat source, and the accumulated geometrical tolerance of all the cladding tracks results in the final tolerance of the finished product. The flatness of a cladding track is a key evaluation indicator of geometrical precision, and lower flatness represents the fluctuations and inconsistencies of cladding track morphology [8,9,10,11]. Furthermore, it has been reported that the lower flatness of a cladding track leads to cracks, pores, etc. more easily and even affects the bonding between the cladding track and substrate [12,13,14,15]. Consequently, it is a significant issue to ensure a higher flatness of a track during the entire cladding process.

A few researchers have revealed that during the laser cladding process, the uneven flow rate and uneven solidification rate of molten materials affect the momentary morphology of a molten pool and its dynamic evolution; furthermore, such momentary morphology determines the flatness of a cladding track [16,17,18,19,20,21]. Tarasankar et al. [22] analyzed the impact of different molten pool aspect ratios on the flatness of a cladding track through numerical simulations and experiments. Their results showed that a higher molten pool aspect ratio promotes a more uniform fluidity and solidification rate of a molten pool, and thus improves the flatness of a cladding track. John et al. [23] and Huang et al. [24] studied the effects of laser power and scanning speed on the fluidity and morphology of a molten pool. Their studies concluded that by controlling process parameters, the morphology of a molten pool can be regulated, and even the flatness of a cladding track can be enhanced. While most recent studies pay attention to the qualitative analysis of flatness, nowadays, quantitative analysis is more desirable for a precision manufacturing process. In summary, in order to control the flatness of a cladding track, it is critical to establish the quantitative relationships between process parameters and flatness.

It is common that in the quantitative analysis process, numerical simulations are given priority relative to experimental methods. To attain a relatively precise numerical simulation of the laser cladding process for flatness statistics, the physical facts—including the flow of molten material, the melting process from solid powder to thermal liquid, the solidifying process from thermal liquid to solid, and the heating and cooling process—must be considered. This type of analysis is performed in the mesoscale. Wang et al. [25] established a mesoscale model, by which they simulated the melting and solidifying processes of metal powder and substrate at different laser powers and scanning speeds. Liu et al. [26] analyzed the stress–strain distribution on the cladding track and the flow velocity distribution of molten material using a mesoscale model. Chen et al. [27] studied the influence of the powder-spray angle and velocity on the molten pool morphology using a mesoscale model. In addition, Huang et al. [28] simulated the heat transfer and energy absorption involving the laser, powder, and substrate surface based on a mesoscale model. In conclusion, it is feasible to use mesoscale models to simulate molten pool evolution and establish the relationship between process parameters and cladding track flatness.

In this study, a parameter window for CoCrMoSi alloys based on the relative cladding track flatness is established via mesoscale numerical simulations. The procedure for establishing this window is illustrated in Figure 1. Firstly, a mesoscale model for the laser cladding process was developed based on computational fluid dynamics (CFDs). Subsequently, simulations of single-track cladding were conducted to analyze the effects of molten pool flow and solidification on the cladding track flatness. An evaluation index for flatness was constructed, and the influences of laser power, scanning speed, and powder feeding rate on flatness were determined through simulations and physical experiments. Finally, a parameter window of flatness was established using simulation and experimental results. This work provides valuable insights for controlling cladding track flatness.

## 2. Experimental and Simulation Schedule

### 2.1. Experimental Procedure

In order to research the influence of processing parameters on the cladding track flatness, a laser cladding experiment was established, as shown in Figure 2. Argon gas was used as a shielding gas to avoid the oxidization of the cladding track during the cladding process.

The base material was 40Cr10Si2Mo with a size of 40 mm × 16 mm × 10 mm, and CoCrMoSi powder with a spherical particle size of 60 μm~150 μm was chosen as the material for laser cladding. The CoCrMoSi powder, produced by Kennametal Stellite, has its chemical composition detailed in Table 1. Firstly, the surface of the substrate was polished with an angle grinder and sandpaper, and an ethanol solution was used to clean the surface. Secondly, the powder was dried in a drying oven at 80 °C for 1 h. The substrate was heated up to 200 °C to carry out laser cladding experiments with different process parameters. Finally, the flatness of the cladding track was analyzed quantitatively.

### 2.2. Simulation Scheme

The study selected laser power, scanning speed, and powder feed rate as the three parameters for investigation. This is because these parameters determine the melt pool morphology, which in turn affects the flatness of the clad layer. Thus, a three-factor and four-level orthogonal test table including three process parameters was designed to study the influence of these parameters on the flatness of the cladding track, as shown in Table 2.

### 2.3. Flatness Evaluation Index

The sample surface was cleaned with a dissolving agent and then dried in a drying oven. Subsequently, the morphology of the cladding track was captured using a three-dimensional shape optical scanning measuring instrument model Cronos3D, which is manufactured by Open Technologies, Italy, as shown in Figure 3. Finally, the surface flatness analysis was carried out by importing the data into MeshLab 2023.12.

To evaluate the straightness of the cladding track, the contour line is extracted along the interface between the substrate and the cladding track. The symbol of “***W_e_***” represents straightness, calculated as the ratio of the average minimum width ***W*_1_** to the average maximum width ***W*_2_** of the cladding track profile, as shown in Figure 3b.

The data at the lowest point on the contour line are extracted and noted as *x*_11_, *x*_12_, *x*_13_, …, *x*_1n_, and the minimum width *W*_1_ is calculated as follows:(1)$$W_{1}=\frac{x_{11}+x_{12}+x_{13}+\cdot \cdot \cdot {+x}_{1n}}{n}$$

The data at the highest point on the contour line are extracted and noted as *x*_21_, *x*_22_, *x*_23_, …, *x*_2n_, and the maximum width *W*_2_ is calculated as follows:(2)$$W_{2}=\frac{x_{21}+x_{22}+x_{23}+\cdot \cdot \cdot {+x}_{2n}}{n}$$

The formula for calculating the degree of straightness *W_e_* is as follows:(3)$$W_{e}=\frac{W_{1}}{W_{2}}$$

The peak-to-valley height measurement was used to evaluate the height fluctuation of the cladding track, and the contour line was intercepted along the center of the cladding track. The symbol “*H_e_*” represents the peak-to-valley height, which is calculated by the ratio of the average minimum height *H*_1_ to the average maximum height *H*_2_ of the cladding track contour, as shown in Figure 3c.

The lowest point data on the contour line are extracted and noted as *d*_11_, *d*_12_, *d*_13_, …, *d*_1*n*_, and the minimum height *H*_1_ is calculated as follows:(4)$$H_{1}=\frac{d_{11}+d_{12}+d_{13}+\cdot \cdot \cdot {+d}_{1n}}{n}$$

The highest point data on the contour line are extracted and noted as *d*_21_, *d*_22_, *d*_23_, …, *d*_2*n*_, and the maximum height *H*_2_ is calculated as follows:(5)$$H_{2}=\frac{d_{21}+d_{22}+d_{23}+\cdot \cdot \cdot {+d}_{2n}}{n}$$

The formula for calculating peak-to-valley height *H_e_* is as follows:(6)$$H_{e}=\frac{H_{1}}{H_{2}}$$

The formula for calculating the degree of flatness *Z_e_* is as follows:(7)$$Z_{e}=\frac{W_{e}{+H}_{e}}{2}$$

## 3. Development of Numerical Models

During the laser cladding process, the laser continuously radiates the powder particles to form a mesoscopic transient molten pool, which undergoes complex phase transition processes, such as melting, evaporation, and solidification, before ultimately forming the cladding track. Mesoscale models considering complex thermophysical phenomena, such as melt flow, solidification, and heat transfer, provide a realistic simulation of molten pool evolution and the impact of process parameters on the flatness of the cladding track. The model employs the Discrete Element Method (DEM) to simulate the flow and distribution of powder, effectively capturing the dynamic behavior of powder particles, including collisions, friction, and flow characteristics, thereby enabling detailed analysis and optimization of the powder source performance. Additionally, it integrates Computational Fluid Dynamics (CFDs) to more accurately predict and control various physical phenomena in the laser cladding process, facilitating the optimization of process parameters and enhancement of cladding track quality.

### 3.1. Thermal Property Parameters of Materials

The powder material used was CoCrMoSi, while the substrate material was 40Cr10Si2Mo. Due to the extensive temperature range reached during cladding processing, it is crucial to consider the thermal performance and temperature of the material in the calculation process. In this study, all thermal properties except for specific heat capacity were calculated by the JMatPro software 11.2 based on the elemental composition of the materials. The changes in density, thermal conductivity, kinematic viscosity, and enthalpy with temperature are depicted in Figure 4.

The melting points of CoCrMoSi and 40Cr10Si2Mo are both approximately 1673.15 K. As shown in Figure 4, there is a distinct transitional region between 1500 K and 1750 K for both materials. This phenomenon is driven by phase transitions resulting from thermodynamic factors and the principle of energy minimization. Within this temperature range, the atomic and crystal structures of both materials gradually change to accommodate the energy variation demands of the external environment. As the temperature increases, atomic thermal motion intensifies, leading to the reorganization and adjustment of chemical bonds within the materials. This transformation is not merely a simple physical process; it involves complex interactions of quantum mechanics and thermodynamics. On the one hand, thermodynamic factors compel the materials to transition toward lower energy states. During this transitional zone, significant changes in thermodynamic parameters, such as entropy and enthalpy, occur, affecting the stability and phase transition behavior of the materials. On the other hand, the principle of energy minimization drives the materials to reduce the system’s total energy by altering their crystal structures and atomic arrangements. Such phase transitions may result in substantial changes to the physical and chemical properties of the materials, including hardness, strength, and electrical conductivity.

### 3.2. Prediction Model of Laser Cladding Morphology

A finite element model was developed to simulate the evolution of cladding track flatness during the processing and the final morphology of the cladding tracks, as shown in Figure 5. During the simulation, the powder and heat sources remained stationary while the substrate moved at a specified velocity along the x-direction. Model boundary conditions involve two main aspects. Firstly, the molten pool driving force theory is applied as the boundary condition for the mesoscale model. Secondly, physical boundary conditions are defined for the entire computational domain. The upper surface is assigned a pressure boundary condition, the surrounding surfaces are set with continuous boundary conditions, and the lower surface is treated as a fixed wall boundary condition.

In the finite element model of the laser cladding, the heat input can be described by the Gaussian heat source model proposed by Carl Friedrich Gauss et al. [29], as in Equation (8). The interaction between the metal powder and the heat source is described by the heat conduction equation and the absorption rate equation as in Equation (9) [30]. The thermal conductivity behavior of the workpiece can be described by the Fourier equation, as in Equation (10) [31]. Thermal convection and radiation between the workpiece and its surroundings can be calculated by Newton’s law and Stefan-Boltzmann’s law, respectively, as in Equation (11) [32]. The geometric morphology equation solves the melt channel morphology during the laser cladding process, as in Equation (12) [33]:(8)$$q\left(r,z\right)=\frac{3P}{\pi r_{0}^{2}h}\exp\left(-\frac{3r^{2}}{r_{0}^{2}}\right)u\left(z\right)$$
where *q*(*r*,*z*) represents the intensity distribution of the heat source, *P* represents the laser power, *r*_0_ represents the effective radius of the heat source, *h* represents the depth of action of the heat source, and *u(z)* represents the unit step function;
(9)$$\left\{\begin{array}{c} \rho C_{p}\left[\frac{\partial T}{t}+\underset{u}{\to }\nabla H\right]=\nabla \cdot \left(k\nabla T\right)-\frac{\partial H}{\partial t}-\rho \underset{u}{\to }\cdot \nabla H\\ \left(\frac{4}{3}\pi r_{p}^{3}\right)\rho_{p}C_{p}^{P}∆T=\eta_{p}\left(\frac{P}{\pi r_{b}^{2}}\right)\left(2\pi r_{p}^{2}\right)t_{f} \end{array}\right.$$
where *ρ* represents the density of the liquid metal, *k* is the thermal conductivity, *C_p_* is the specific heat capacity, *H* is the enthalpy of the melt pool, *t* is the flow time of the melt pool, *η_p_* is the energy absorption rate of the powder particles, *r_b_* is the radius of the laser beam, *r_p_* is the radius of the powder particles, $\rho_{p}$ is the powder density, $C_{p}^{P}$ is the specific heat capacity of the powder particles, *P* is the laser power, and *t_f_* is the flight time of the powder particles in the laser beam;
(10)q=−k∇T
where *T* represents temperature and *k* is thermal conductivity varying with temperature;
(11)$$\left\{\begin{array}{c} q_{conv}=h_{conv}\left(T-T_{env}\right)\\ q_{rad}=\varepsilon_{rad}\sigma_{rad}\left(T^{4}-T_{env}^{4}\right) \end{array}\right.$$
where *T_env_* represents the environment temperature, *h_conv_* is thermal convection coefficient, *ε_rad_* is thermal emissivity, and *σ_rad_* is Stefan-Boltzmann constant;
(12)$$h(x)=A\cdot exp(\frac{{\left(x-x_{0}\right)}^{2}+{\left(y-y_{0}\right)}^{2}}{2\sigma^{2}})\cdot (1-exp(-\alpha T))+B{\left(\frac{\partial T}{\partial Z}\right)}^{n}$$
where *h*(*x*) represents the solidification behavior of molten metal during the cladding process, *A* is the laser density, (*x_0_*, y*_0_*) is the position of the center of the laser beam, α is the material absorption coefficient, σ is the radius of the laser beam, *t* is the time, *B* is a constant, $\frac{\partial T}{\partial Z}$ is the temperature gradient, and *n* is the solidification rate.

### 3.3. Parametric Evaluation of Flatness

The critical point data on the cladding track were extracted for quantitative analysis of the flatness during the cladding process. The critical point data extraction is shown in Figure 6. The critical points were extracted to identify the location distribution of different points, and the maximum and minimum points were further determined. Based on the distribution of the maximum and minimum points, the straightness and peak-to-valley height were calculated to partition the flatness level and the processing parameter windows were further constructed.

## 4. Mechanism of Cladding Layer Flatness Formation

The flatness of a cladding layer primarily arises from liquid metal flow and the solidification of the cladding process. Figure 7 depicts the solid domain and the fluid domain during the cladding process. In the liquid metal flow phase, low flatness was caused by uneven distribution fluidity on the cladding track. The uneven distribution of fluidity can be attributed to the non-uniform flow rates within the fluid domain, resulting in differences in the surface tension of the liquid metal. Consequently, in solidification, uneven cooling can lead to surface irregularities, further affecting flatness.

### 4.1. Effect of Molten Pool Velocity Field on Flatness

Figure 8 illustrates the velocity vector distribution in the molten pool, and the arrow’s direction represents the direction of fluid movement. The movement of molten metal directly influences the macroscopic morphology of the cladding track. High flow velocity regions typically accompany effective momentum transfer, facilitating uniform material diffusion and thereby forming a flat surface. In contrast, low flow velocity regions are often linked to reduced momentum transfer and potential local solidification, which can lead to surface irregularities.

At *t* = 0.1 s, the laser directly heats the substrate, forming a small, smooth-edged circular molten pool. At this point, the powder feed has not yet begun, resulting in a relatively uniform velocity distribution within the molten pool and a smooth surface. At *t* = 0.3 s, continuous laser heating causes the molten pool to expand in size, and the powder feed starts. The metal powder constantly falls into the molten pool and melts, which causes the velocity field of the molten pool to fluctuate. At *t* = 0.6 s, due to the different densities of powder and substrate material, two velocity fields in different directions are formed in the melt pool, thus accelerating the fusion of powder and substrate. At *t* = 1.5 s and *t* = 2.0 s, velocity vector concentration areas appear at the molten pool edges, overlapping with the surface irregularities, indicating that edge-concentrated velocity fields cause surface roughness. At *t* = 3.0 s, the laser heating ceases, and the clad layer enters the cooling phase. The velocity field begins to even out, but the surface irregularities have already formed.

### 4.2. Effect of Solidification Behavior on Flatness

The solidification process is a critical stage in the formation of the cladding track, and the solidification first occurs at the edge of the molten pool. Figure 9 shows the solid domain around the molten pool.

At *t* = 0.1 s, the laser directly heats the substrate to form a molten pool, and the edge of the solidification domain is smooth. At *t* = 0.3 s, the powder feeding begins. Under the intense influence of the laser, some powder particles are splashed and scattered onto the substrate surface due to the recoil force of the molten pool, resulting in irregular solidification at the molten pool edges. Between *t* =0.6 s and 1.5 s, the molten pool size increases due to thermal accumulation. This enlargement also affects the solidification region at the edge of the molten pool, worsening the irregular solidification at the edge. At *t* = 2.0 s, the molten pool size ceases to increase, and the temperature distribution of the molten pool stabilizes. At this point, the flatness of the cladding track is determined. At *t* = 3.0 s, powder feeding and laser heating stop, and the contour of the cladding track is formed. In general, the uneven distribution of temperature in the molten pool leads to irregular solidification at the edge of the molten pool, thus reducing the flatness.

### 4.3. Experimental Validation

To validate the accuracy of the simulation, a laser cladding experimental platform was established, as shown in Figure 10. Argon gas was used as a shielding gas during the cladding process to prevent oxidation. A single-track cladding process was performed with a laser power of 2400 W, a scanning speed of 280 mm/min, and a powder feed rate of 5 g/min, achieving a track length of 35 mm. Subsequently, the track cross-sections were etched using a corrosion solution consisting of 22 mL of HNO_3_ and 6 mL of HCl. The dimensions of the melt pool were observed and measured using the Nikon MM400LM measurement microscope.

A simulation of single-track cladding was conducted using identical process parameters as in the experiment. The molten pool width and depth were measured at locations along the molten track in the ZY section, as shown in Figure 11a. The red area in the ZY section indicates the area above the melting temperature where the liquid pool is formed. The simulation results indicate that the molten pool width is 5.75 mm, and the depth is 3.245 mm, as shown in Figure 11b. The experimental and simulated molten pool sizes are compared, as shown in Figure 12. The experimental results show that the molten pool has a width of 5.53 mm and a depth of 3.136 mm. Based on the calculation of the difference between the experimental and simulated melt pool height and width, and dividing it by the experimental melt pool dimensions, the resulting error is 0.03968. The error between the experimental and simulated values is less than 4. Thus, the proposed numerical model can accurately predict the molten pool morphology.

## 5. Relationship between Process Parameters and Flatness

### 5.1. Influence of Process Parameters on Flatness

In Figure 13, the cladding track morphology at a laser power of 2100 W is shown under different scanning speeds and powder feeding rates. Using MeshLab, the contour and key points of the cladding track were extracted to calculate the flatness of the cladding tracks according to Equations (1)–(7). The flatness values under different conditions are calculated as 58.62%, 59.64%, 64.21%, and 72.55%, respectively. These results indicate that increasing scanning speed and powder feed rate significantly improves flatness and reduces jagged structures. Increasing the scanning speed reduces the laser dwell time on the substrate surface, thereby shortening the melt pool formation time. The reduced dwell time limits heat accumulation, resulting in a smaller molten pool size. This leads to a more uniform solidification process, reducing the impact of thermal diffusion and molten pool fluidity on the surface morphology of the cladding track, thereby enhancing the cladding track flatness. Simultaneously, increasing the powder feed rate supplies more cladding material, further reducing the molten pool size and accelerating the cooling rate. Faster solidification minimizes molten pool fluctuations during solidification, improving surface flatness. Additionally, the increased powder material helps fill minor defects in the track, leading to a smoother surface. In summary, increasing the scanning speed and powder feed rate leads to a decrease in molten pool size and an acceleration in solidification rate, which collectively contribute to improvements in track flatness. These factors significantly reduce the formation of jagged structures and enhance the overall surface quality of the cladding track.

In Figure 14, the cladding track morphology at a laser power of 2200 W is shown under different scanning speeds and powder feeding rates. The flatness of the cladding track under various conditions was calculated to be 57.42%, 59.27%, 72.37%, and 71.32%, respectively. It is evident that a relatively smooth cladding track was formed at a scanning speed of 220 mm/min and a powder feeding rate of 5.5 g/min. This favorable result is attributed to the combination of scanning speed and powder feeding rate, which ensures that the molten pool has sufficient size and solidification time, thus reducing surface defects while maintaining the flatness of the cladding track. When the scanning speed was increased to 240 mm/min and the powder feeding rate was set at 5 g/min, the cladding track still maintained good flatness. Although the scanning speed increased, the relatively low powder feed rate kept the molten pool size manageable, preventing excessive powder accumulation. However, further increasing the scanning speed to over 260 mm/min and the powder feeding rate to over 6 g/min resulted in excessive powder deposition and insufficient melting time. In this case, the excess powder failed to fully melt, leading to uneven surface formation in the cladding track and a reduction in track flatness. In summary, different scanning speeds and powder feeding rates significantly impact the flatness of the cladding track. A reasonable combination of parameters can effectively improve the surface quality of the cladding track, while excessively high scanning speeds and powder feeding rates result in a decline in flatness.

Figure 15 illustrates the cladding track morphology at laser powers of 2300 W and 2400 W under various scanning speeds and powder feeding rates. The flatness of the cladding track was calculated using formulas, resulting in values of 86.21%, 87.62%, 79.77%, and 77.42% (for 2300 W), and 83.17%, 84.63%, 85.72%, and 91.83% (for 2400 W). The results indicate that the flatness of the cladding track is significantly better at laser powers greater than 2300 W compared to those less than 2200 W. Further analysis shows that higher laser power increases the thickness of the cladding track, but this also results in a decrease in flatness. This phenomenon is attributed to the fact that, while the depth of material melting and the size of the molten pool are increased by higher laser power, the fluidity of the molten pool is also enhanced, leading to more irregular surface structures. To address this issue, the impact of different scanning speeds on the thickness and flatness of the cladding track was analyzed. Increased scanning speed can reduce the thickness of the cladding track and improve flatness by shortening the laser dwell time on the substrate surface, thereby reducing irregularities caused by excessive heating of the molten pool. However, excessively high scanning speeds lead to inadequate heat input, which results in irregular melt pool shapes and diminished flatness. Therefore, optimal scanning speed selection is crucial for improving the surface morphology of the cladding track. Powder feeding rate also plays a critical role in the laser cladding process. Variations in powder feeding rate revealed that higher rates increase the thickness of the cladding track and decrease flatness. This results from excessive powder deposition on the substrate, which surpasses the melting capacity of the melt pool, causing surface fluctuations and irregularities. In contrast, lower powder feeding rates produce thinner and smoother cladding tracks. However, it is essential to balance the powder supply with cladding layer thickness to ensure sufficient mechanical properties. In summary, thorough simulation analysis reveals that different combinations of laser power, scanning speed, and powder feeding rate significantly impact the morphology of the cladding layer. Optimal parameter selection effectively balances thickness and flatness, thereby improving the overall quality of the cladding track.

In order to gain a deeper understanding of the impact of laser power, scanning speed, and powder feeding rate on the flatness of the cladding track, an analysis was conducted. Based on an orthogonal experimental design, two-factor response surfaces were established to evaluate the effects of powder feeding rate versus scanning speed, laser power versus powder feeding rate, and laser power versus scanning speed on flatness, as shown in Figure 16. These response surfaces are essential for uncovering the interactions between factors, facilitating a clearer understanding of how different factor combinations influence the flatness of the cladding track. Moreover, they streamline the analysis of multi-factor relationships and provide a theoretical foundation for optimizing laser cladding process parameters. As shown in Figure 16a, a higher laser power combined with a higher powder feed tends to reduce flatness. Better flatness was observed under the processing parameters of 2300 W and 5 g/min. It can be observed that worse flatness was distributed in the higher laser power and the slower scanning speed shown in Figure 16b. At conditions of 2300 W and 260 mm/min, better flatness can be found. Figure 16c shows the phenomenon that higher scanning speeds and lower powder feeds generally improve flatness. The optimum conditions correspond to a scanning speed of 260 mm/min and a powder feed rate of 5 g/min. In conclusion, it is crucial to balance parameters. The best flatness of the cladding track can be formed at conditions of moderate laser power and lower powder feed combined with higher scanning speeds.

Based on the flatness calculations of Formulas (1)–(7) established in this study, flatness calculations were conducted on the experimental and simulated melting trajectories obtained from 16 different parameter sets and the results were compared, as illustrated in Figure 17. The trends of the flatness curves obtained from both the experiments and simulations were generally consistent, indicating that the simulation method achieved a high level of accuracy in capturing the trend of flatness changes. However, disparities were noted between the experimental and simulated values for specific parameter combinations, with the most significant variance observed in the 16th parameter set. The absolute error of the maximum deviation was calculated to be less than 6%. Therefore, it can be concluded that the simulation results based on this model effectively predict the flatness of the laser melting trajectory.

### 5.2. Window of Processing Parameters

Based on the impact of different laser powers, scanning speeds, and powder feeding speeds on the flatness, the evaluation interval was established. With low scanning speed (*v* < 260 mm/min), low laser power (*P* < 2300 W), and high powder feed rate (*P_f_* > 5.5 g/min), the flatness was low, leading to the existence of a zigzag structure on the surface of the cladding track. The reason for this phenomenon is that the low scanning speed, low laser power, and high powder feeding rate lead to a decrease in the temperature of the molten pool, and that a large number of metal powders enter the molten pool and are partially melted. The unmelted powder leads to an unstable velocity field in the molten pool, which results in uneven solidification and the formation of multiple zigzag structures on the surface of the cladding track. With high scanning speed (*v* > 260 mm/min), high laser power (*P* > 2300 W), and low powder feed rate (*P_f_* < 5.5 g/min), the flatness is high, and the surface of the fused cladding track is smooth, without noticeable structural bumps. This is due to the fact that the high laser power enhances the temperature of the molten pool, the low powder feed rate results in a uniform distribution of the metal powder in the molten pool, and the high scanning speed ensures a uniform solidification of the molten pool, thus guaranteeing a high flatness of the cladding track. Under the conditions of a scanning speed of 280 mm/min, a laser power of 2400 W, and a powder feeding rate of 5 g/min, the cladding track achieved the highest flatness. This parameter combination not only significantly enhances the surface quality of the cladding track, but also provides a critical optimization basis for the practical application of laser cladding processes. Achieving optimal flatness with these parameters effectively reduces the need for post-processing, improves the structural performance and durability of the final product, and has substantial implications for the application of laser cladding technology in the manufacturing industry. To visually display the relationships between the flatness of the cladding track and different processing parameters, a parameter window was established, as shown in Figure 18. Selecting process parameters based on this window helps achieve optimal forming quality and provides suitable parameter ranges for further research.

## 6. Conclusions

In this paper, a mesoscale model of CoCrMoSi alloy single-pass cladding was established to simulate the single-pass cladding forming process. The formation mechanism corresponding to the flatness of the cladding layer was revealed, and the impacts of laser power, scanning speed, and powder feeding rate on the flatness of the cladding layer were analyzed. Based on the simulation results, a parameter window for the relationships between the flatness of the cladding layer and the processing parameters was established. The conclusions can be summarized as follows:(1)An evaluation index of flatness is proposed, and the quantitative analysis of the flatness of the cladding layer is achieved. The size of the flatness of the cladding layer can be accurately evaluated based on the acquired data of the cladding layer from experiments and simulations.(2)The mechanism leading to the unevenness of the cladding layer was revealed via single-pass cladding simulation. The unevenness of the cladding layer was attributed to the mismatch of processing parameters, which led to an uneven velocity field in the melt pool. This uneven velocity field caused irregular solidification of the melt pool. Ultimately, a jagged structure appeared on the surface of the cladding layer, reducing its flatness.(3)At low scanning speed (*v* < 260 mm/min), low laser power (*P* < 2300 W), and high powder feeding rate (*P_f_* > 5.5 g/min), the surface flatness of the cladding track was low, resulting in the presence of jagged structures on the surface of the cladding track. At high scanning speed (*v* > 260 mm/min), high laser power (*P* > 2300 W), and low powder feeding rate (*P_f_* < 5.5 g/min), the surface flatness of the cladding track was high, and there were no obvious concave or convex structures.

## Figures and Tables

**Figure 1 materials-17-05225-f001:**
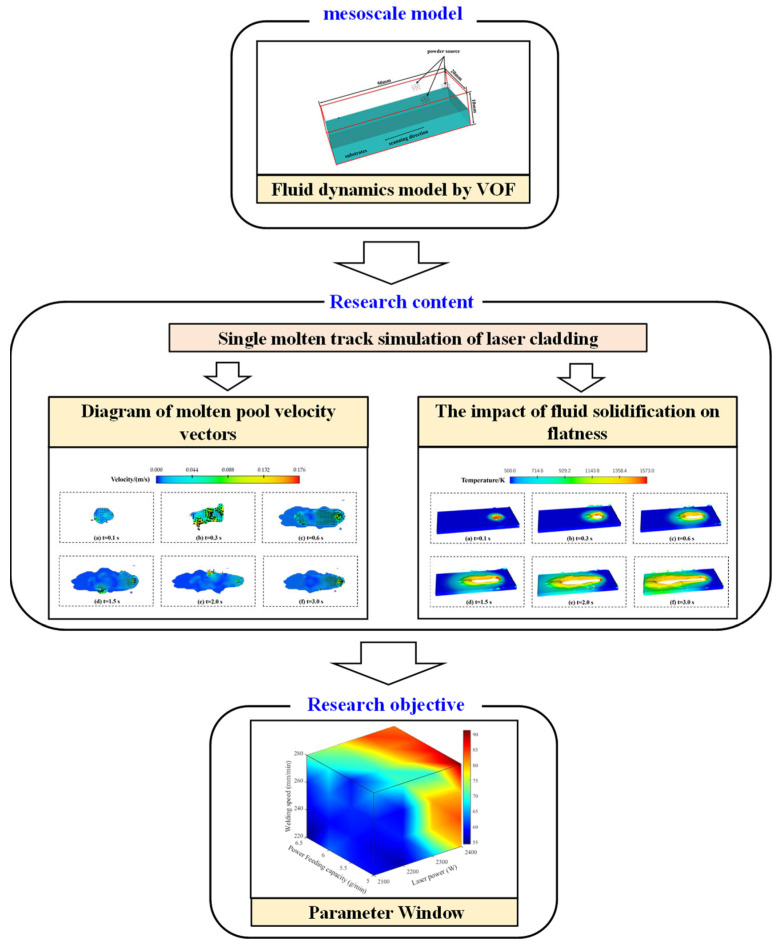
The method of obtaining the parameter window.

**Figure 2 materials-17-05225-f002:**
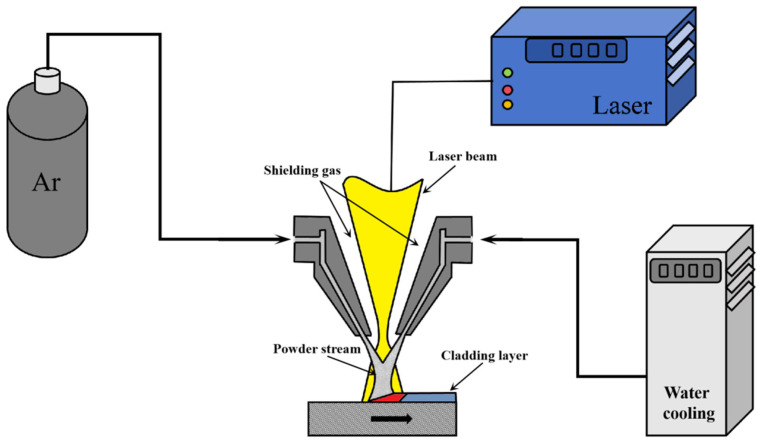
Laser cladding coaxial-powder feeding platform schematic diagram.

**Figure 3 materials-17-05225-f003:**
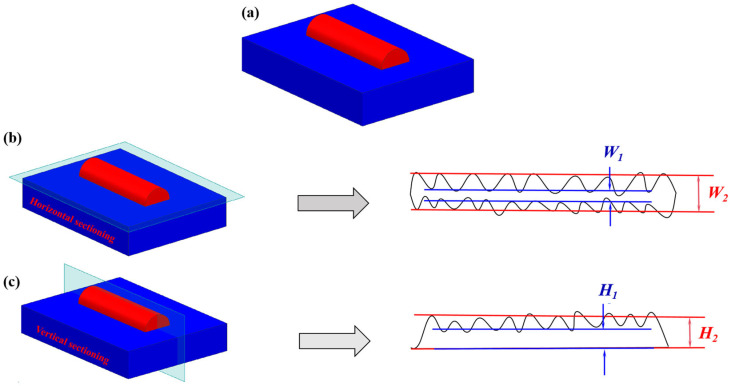
Flatness analysis of the cladding track (**a**) the morphology of a cladding track, (**b**) Straightness evaluation, (**c**) Peak-to-Valley Height evaluation.

**Figure 4 materials-17-05225-f004:**
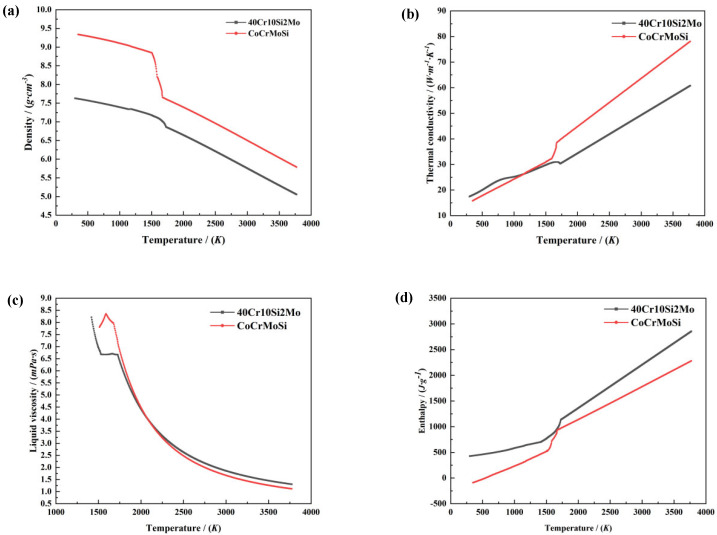
Variation of thermophysical parameters with temperature. (**a**) Density; (**b**) thermal conductivity; (**c**) dynamic viscosity; (**d**) enthalpy.

**Figure 5 materials-17-05225-f005:**
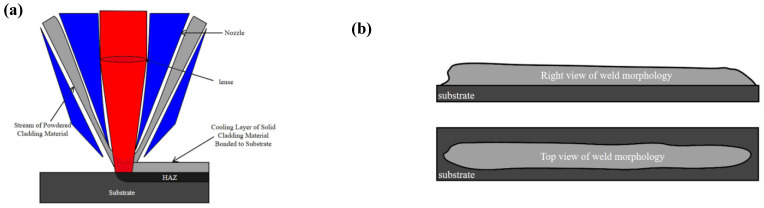
Laser cladding process diagram (**a**); cladding layer topography and geometry (**b**).

**Figure 6 materials-17-05225-f006:**
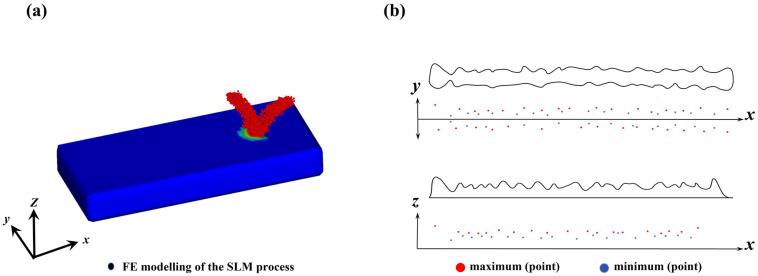
Finite element model of coaxial powder feeding (**a**); Extraction key point data of cladding layer contour (**b**).

**Figure 7 materials-17-05225-f007:**
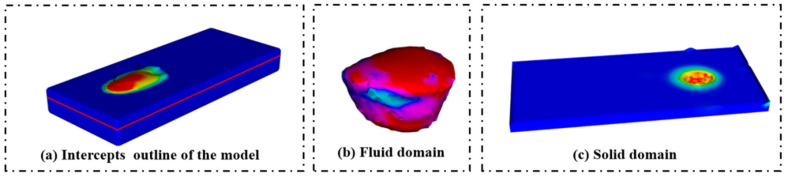
Schematic diagram of cladding layer flatness: (**a**) intercepts outline of the model; (**b**) solid domain; (**c**) fluid domain.

**Figure 8 materials-17-05225-f008:**
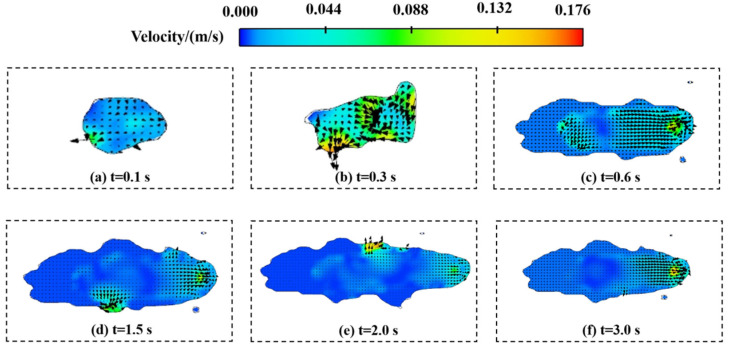
Molten pool velocity vector diagram.

**Figure 9 materials-17-05225-f009:**
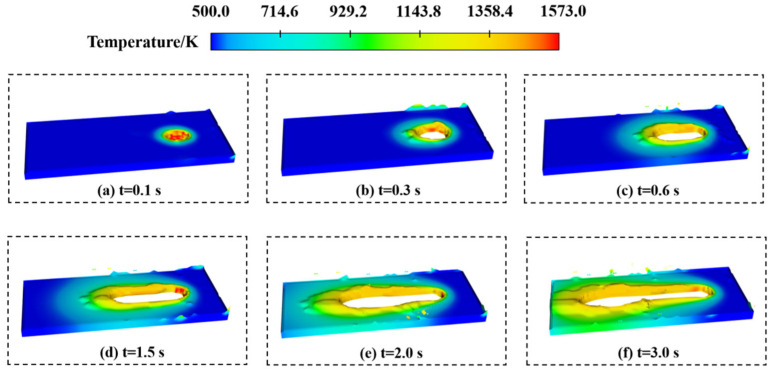
The solidification process of the solid domain edges the molten pool.

**Figure 10 materials-17-05225-f010:**
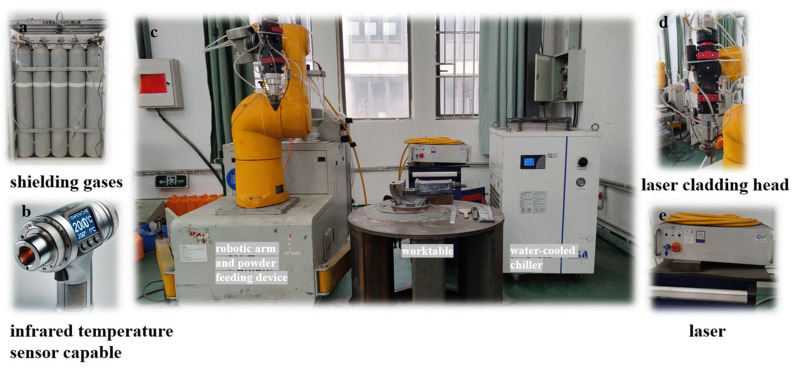
Laser powder bed cladding working platform. (**a**) shielding gases; (**b**) infrared temperature sensor capable; (**c**) robotic arm and powder feeding device, worktable and water-cooled chiller; (**d**) laser cladding head; (**e**) laser.

**Figure 11 materials-17-05225-f011:**
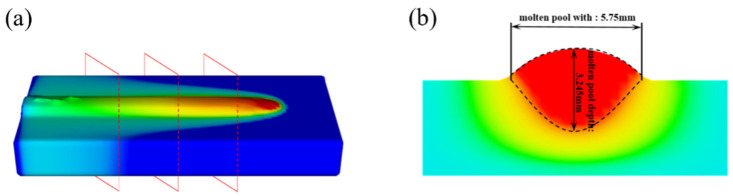
Simulated molten pool size: (**a**) three-dimensional schematic, (**b**) ZY cross-section.

**Figure 12 materials-17-05225-f012:**
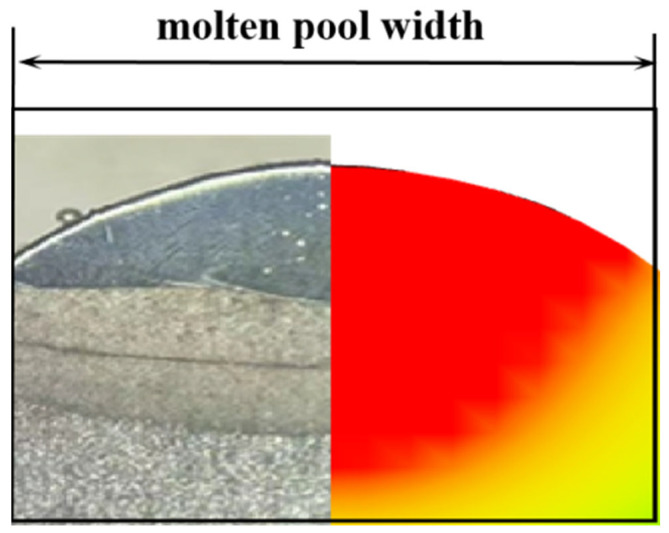
Experimental versus simulated molten pool size comparison.

**Figure 13 materials-17-05225-f013:**
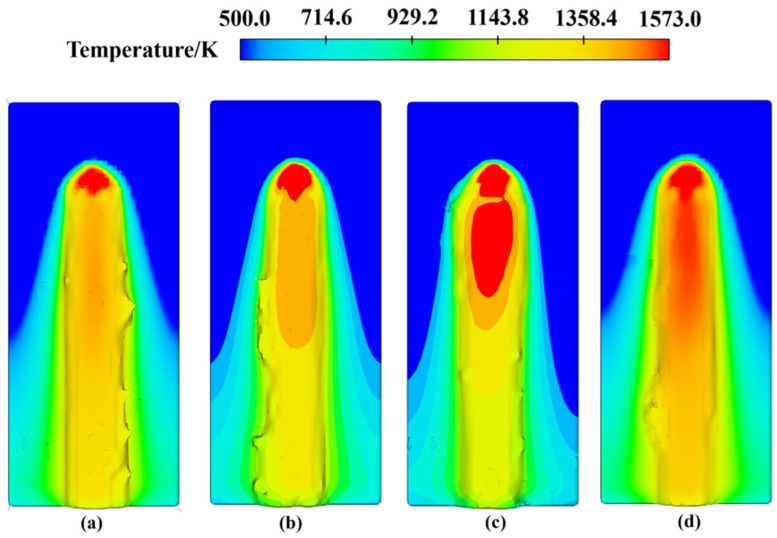
Morphology of cladding track under 2100 W under different conditions: (**a**) 220 mm/min, 5 g/min; (**b**) 240 mm/min, 5.5 g/min; (**c**) 260 mm/min, 6 g/min; (**d**) 280 mm/min, 6.5 g/min.

**Figure 14 materials-17-05225-f014:**
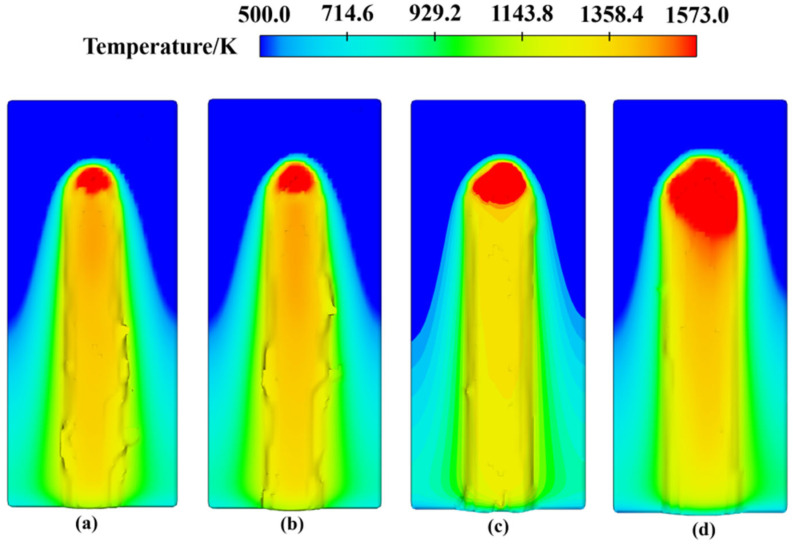
Morphology of cladding track under 2200 W under different conditions: (**a**) 220 mm/min, 5.5 g/min; (**b**) 240 mm/min, 5 g/min; (**c**) 260 mm/min, 6.5 g/min; (**d**) 280 mm/min, 6 g/min.

**Figure 15 materials-17-05225-f015:**
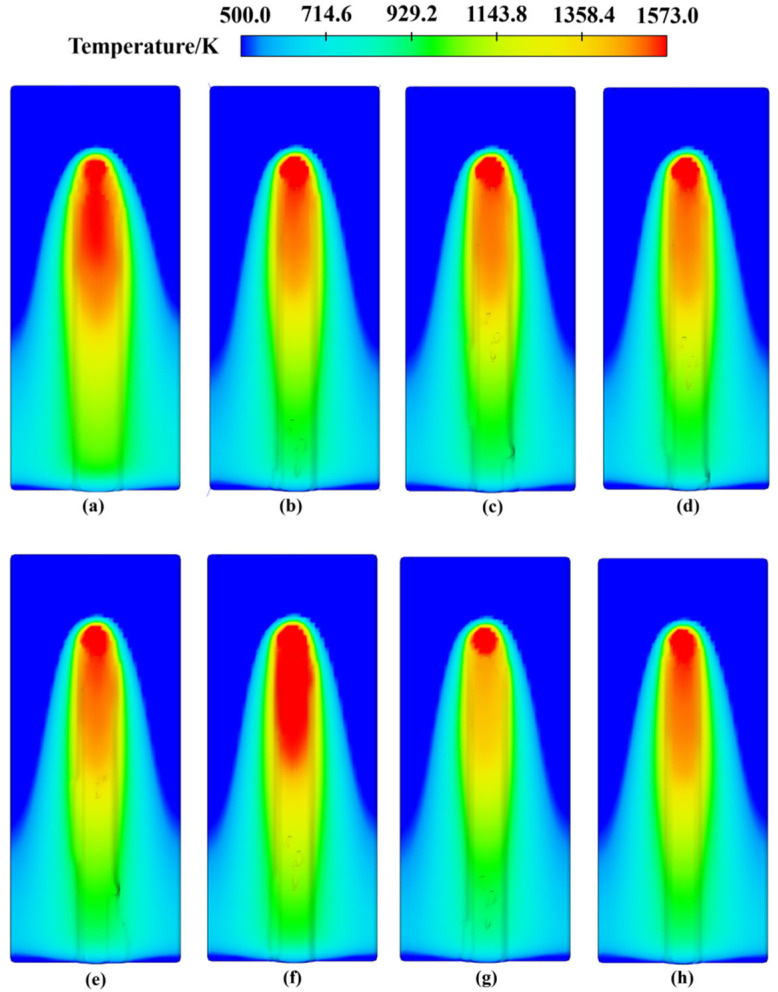
Morphology of cladding track (**a**–**d**) under 2300 W and (**e**–**h**) 2400 W under the following conditions: (**a**) 220 mm/min, 6 g/min; (**b**) 240 mm/min, 6.5 g/min; (**c**) 260 mm/min, 5 g/min; (**d**) 280 mm/min, 5.5 g/min; (**e**) 220 mm/min, 6.5 g/min; (**f**) 240 mm/min, 6 g/min; (**g**) 260 mm/min, 5.5 g/min; (**h**) 280 mm/min, 5 g/min.

**Figure 16 materials-17-05225-f016:**
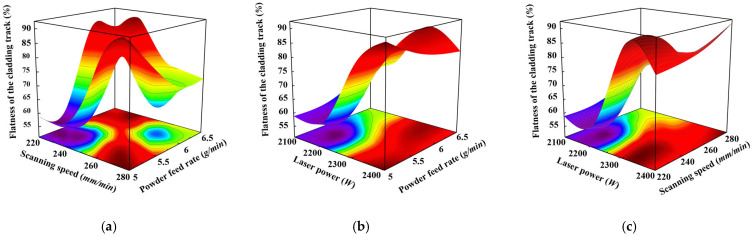
Process parameters and flatness response surfaces: (**a**) Laser power and powder feed rate; (**b**) Laser power and scanning speed; (**c**) Scanning speed and powder feed rate.

**Figure 17 materials-17-05225-f017:**
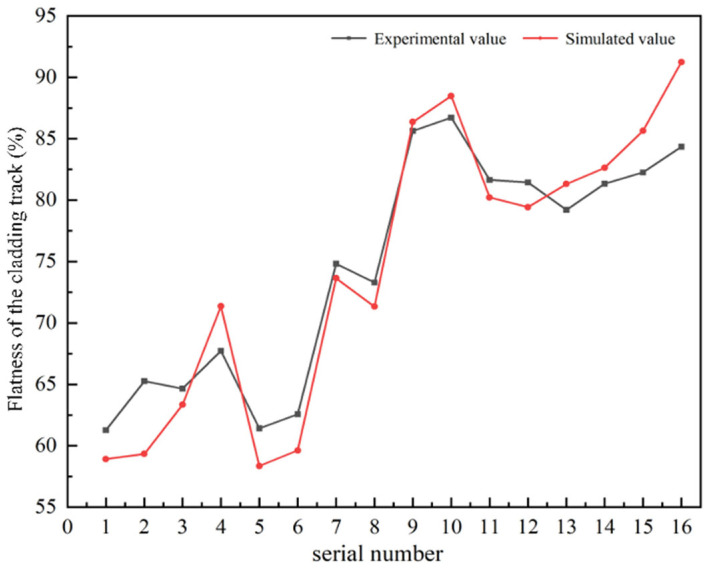
Comparison of test and simulated cladding track flatness.

**Figure 18 materials-17-05225-f018:**
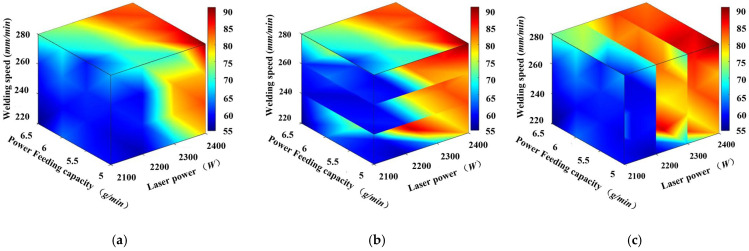
Schematic diagram of the influence of process parameters on flatness (**a**); Horizontal section (**b**); Vertical section (**c**).

**Table 1 materials-17-05225-t001:** Chemical composition of 40Cr10Si2Mo substrate and CoCrMoSi powder. (Mass fraction/%).

Element	C	Mo	Ni	P	Mn	S	Cr	Si	B	O	Fe	Co
40Cr10Si2Mo	0.4	0.76	0.16	0.02	0.3	0.003	9.76	2.33	-	-	Bal.	-
CoCrMoSi	0.01	29.9	0.1	-	-	-	9.8	2.89	0.001	0.005	0.11	Bal.

**Table 2 materials-17-05225-t002:** Experimental scheme for single-pass cladding of CoCrMoSi powder.

Experiment Number	Laser Power (W)	Welding Speed (mm/min)	Powder Feeding Capacity (g/min)
1	2100	220	5
2	2100	240	5.5
3	2100	260	6
4	2100	280	6.5
5	2200	220	5.5
6	2200	240	5
7	2200	260	6.5
8	2200	280	6
9	2300	220	6
10	2300	240	6.5
11	2300	260	5
12	2300	280	5.5
13	2400	220	6.5
14	2400	240	6
15	2400	260	5.5
16	2400	280	5

## Data Availability

The original contributions presented in the study are included in the article, further inquiries can be directed to the corresponding author.

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
