# Peer review of "Influence of Process Parameters on Flatness During Single-Track Laser Cladding"

_materials, 2024, doi:10.3390/ma17215225_

Round 1

Reviewer 1 Report

Comments and Suggestions for Authors

This paper investigates the effect of laser cladding parameters on the cladding track flatness. It presents simulation and experimental results, developing a parameter window for the most optimal properties. The following comments may help improve the quality of the article after a major review of the article, which is my recommendation for this publication. 

Suggestions for improving the article:

1.     Format the article according to Materials journal guidelines.

2.     Develop the purpose of the research presented in the Introduction why you study the CoCrMoSi alloy, why it is on the 40Cr10Si2Mo substrate, why your research is important, what is new/different than in other articles you cite, etc.

3.     2.1 Experimental procedure: what is the chemical composition of the CoCrMoSi powder, who is the manufacturer of it, please specify more information about it. Move this information from 3.1 to 2.1.

4.     What was the methodology for polishing the samples? It was done by hand or automatically? What is the repeatability of surface preparation? In the reviewer's opinion, this is crucial when assessing flatness for laser cladding.

5.     Please specify the name/manufacturer of the 3D optical profiler and the name of the specialized software for surface flatness analysis.

6.     Figure 3 – make the annotation larger to improve the readability of it.

7.     ‘The symbol ‘He’ represents the Peak-to-Valley Height, which is calculated by the ratio of the average minimum height H1 to the average maximum height H2 of the cladding track contour, as shown in Figure (c).’ – should be Figure 3c

8.     Comment on all formulas – please add the number of each formula on the right side of the page in brackets (1), etc. – add this number for the formula W1, W2, We, H1, H2, He, Ze.

9.     Correct the position of the H2 formula, it should be after this: “ the maximum height H2 is calculated by the following formula:”.

10.  Table 2 – specify whether this chemical composition is from your measurement – in that case, add the device and information about it, or this is information from the manufacturer, in this case, add the literature data.

11.  Correct the Figure 4, 16 and 18 to increase the readability of the figure. Increase the label size and remove the dotted lines around each image.

12.  Add more information on the finite element model. What are the types of elements, and what boundary conditions are used?

13.  Verify all the equations for the description so that all of the symbols are defined, for example, what P stands for in (2)

14.  Why are the formula (5) and 𝛼, 𝜎, 𝜕𝑇/𝜕𝑧 are highlighted?

15.  Where is the “B” in the equation (5)?

16.  The arrows in Figure 8 are very small, so it is hard to notice the direction. Consider changing it for better readability of the image.

17.  4.3 Experimental validation: specify the name/manufacturer of the stereo microscope.

18.  What do the different colours from blue to red mean in Figures 11 – 16? Add the legend.

19.  “At conditions of 2300W and 260 mm/min, better flatness can be found. Figure (c) shows the phenomenon that higher scanning speeds and lower powder feeds generally improve flatness.” Like in suggestion nr. 3 – Figure (c), the number is missing.

Author Response

Dear Editors and Reviewers.

Thank you for your letter and for the reviewers’ comments concerning our manuscript entitled “Influence of Process Parameters on Flatness During Single-Track Laser Cladding” (ID: materials-3139208). Those comments are all valuable and very helpful for revising and improving our paper, as well as the important guiding significance to our researches. We have studied comments carefully and have made corrections which we hope to meet with approval. Revised portion are marked in red in the paper. The main corrections in the paper and the responds to the reviewer’s comments are as flowing:

Responds to the reviewer#1’s comments:

  1. Comment:-- Format the article according to Materials journal guidelines.

Response: According to the reviewer's good instruction, we have revised the whole manuscript carefully. In addition, we have requested several experienced English-language paper writers to review the typesetting of our papers. We hope that this layout will be acceptable for the next review process..

  1. Comment: - Develop the purpose of the research presented in the Introduction why you study the CoCrMoSi alloy, why it is on the 40Cr10Si2Mo substrate, why your research is important, what is new/different than in other articles you cite, etc.

Response: Thanks for reviewer’s good suggestions. Why are you studying the CoCrMoSi alloy and its use on the 40Cr10Si2Mo substrate? And the content has been added in our revised manuscript, the added content is as following:

The 40Cr10Si2Mo alloy is a material that is widely used in the manufacture of valves for low-speed engines. However, as the power of low-speed engines increases, the operating conditions of these valves are becoming more demanding [5-6]. The current method of controlling grain size to improve valve performance through the electroforming-forging process is no longer sufficient to meet these stringent requirements [7]. Therefore, the use of laser cladding of CoCrMoSi iron-based high-temperature alloys on valve surfaces can effectively improve valve performance. 

  1. 3. Comment: 1 Experimental procedure: what is the chemical composition of the CoCrMoSi powder, who is the manufacturer of it, please specify more information about it. Move this information from 3.1 to 2.1.

Response: Thanks for reviewer’s good suggestions. It is really true as the reviewer said. Based on the valuable comments from the reviewers, section 3.1 has been moved to section 2.1 in this revised version, and the manufacturers of the alloy powders have been added as following:

The CoCrMoSi powder, produced by Kennametal Stellite, has its chemical composition detailed in Table 2. 

  1. Comment: -What was the methodology for polishing the samples? It was done by hand or automatically? What is the repeatability of surface preparation? In the reviewer's opinion, this is crucial when assessing flatness for laser cladding.

Response: Thank you for the reviewer's suggestions. In response to the valuable comments, our answer to the question regarding how the samples are polished and the reproducibility of the surface treatment is as follows:

The samples were polished manually. We carefully controlled the operating conditions during the polishing process to ensure reproducibility. To verify the reproducibility of the polishing process, we conducted multiple polishing experiments on different sample sets and recorded detailed data for each experiment. The results showed that the surface quality and polishing effects of the samples remained consistent across all experiments. This demonstrates the stability and reproducibility of our polishing method.

  1. Comment:Please specify the name/manufacturer of the 3D optical profiler and the name of the specialized software for surface flatness analysis.

Response: Thanks for reviewer’s suggestions. According to reviewer’s valuable       comments, we have included the experimental apparatus and the software used in the     revised manuscript.

Three-dimensional shape optical scanning measuring instrument, which is manufactured by Open Technologies, Italy, and is model Cronos3D, as shown in Figure 3. Finally, the surface flatness analysis is carried out by importing the data into MeshLab 2023.12.

  1. Comment:Comments from the reviewers on the article's images and some           formatting issuesResponse.

Response: Thanks for reviewer’s suggestions. According to reviewer’s valuable comments,we have made the following changes to the images and content of the article.

Original:( Figure 3–make the annotation larger to improve the readability of it.)  

Figure 3. Flatness analysis of the cladding track (a) the morphology

 of a cladding track, (b) Straightness evaluation, (c) Peak-to-Valley Height evaluation

Modified

Figure 3. Flatness analysis of the cladding track (a) the morphology

 of a cladding track, (b) Straightness evaluation, (c) Peak-to-Valley Height evaluation

Original: The symbol " He " represents the Peak-to-Valley Height, which is calculated by the ratio of the average minimum height H1 to the average maximum height H2 of the cladding track contour, as shown in Figure (c).

Modified: The symbol " He " represents the Peak-to-Valley Height, which is calculated by the ratio of the average minimum height H1 to the average maximum height H2 of the cladding track contour, as shown in Figure 3(c).

Original

Modified

(1)

(2)

(3)

(4)

(5)

(6)

(7)

Original: The data of the highest point on the contour line is extracted and noted as d1, d2, d3 ---dn and the maximum height H2 is calculated by the following formula:

Modified: The data of the highest point on the contour line is extracted and noted as d1, d2, d3 ---dn and the maximum height H2 is calculated as follows:

Original

Figure 4. Variation of thermophysical parameters with temperature. (a) Density; (b) thermal conductivity; (c) dynamic viscosity; (d) enthalpy.

Figure 16. Process parameters and flatness response surfaces: (a) Laser power and powder feed rate; (b) Laser power and scanning speed; (c) Scanning speed and powder feed rate.

Figure 18. Schematic diagram of the influence of process parameters on flatness (a); Horizontal section (b); Vertical section (c).

Modified:

Figure 4. Variation of thermophysical parameters with temperature. (a) Density; (b) thermal conductivity; (c) dynamic viscosity; (d) enthalpy.

Figure 16. Process parameters and flatness response surfaces: (a) Laser power and powder feed rate; (b) Laser power and scanning speed; (c) Scanning speed and powder feed rate.

Figure 18. Schematic diagram of the influence of process parameters on flatness (a); Horizontal section (b); Vertical section (c).

Original

Figure 13. Morphology of cladding track under 2100W with different conditions of: (a) 220 mm/min, 5g/min; (b) 240 mm/min, 5.5g/min; (c) 260 mm/min, 6 g/min; (d) 280 mm/min, 6.5g/min

Figure 14. Morphology of cladding track under 2200W with different conditions of: (a)220 mm/min, 5.5g/min; (b) 240 mm/min, 5g/min; (c) 260 mm/min, 6.5g/min; (d) 280 mm/min, 6g/min

Figure 15. Morphology of cladding track (a) to (d) under 2300W and (e) to (h) 2400W at conditions of: (a)220 mm/min, 6g/min; (b) 240 mm/min, 6.5g/min; (c) 260 mm/min, 5g/min; (d) 280 mm/min, 5.5g/min; (e)220 mm/min, 6.5g/min; (f) 240 mm/min, 6g/min; (g) 260 mm/min, 5.5g/min; (h) 280 mm/min, 5g/min

Modified:

Figure 13. Morphology of cladding track under 2100W with different conditions of: (a) 220 mm/min, 5g/min; (b) 240 mm/min, 5.5g/min; (c) 260 mm/min, 6 g/min; (d) 280 mm/min, 6.5g/min

Figure 14. Morphology of cladding track under 2200W with different conditions of: (a)220 mm/min, 5.5g/min; (b) 240 mm/min, 5g/min; (c) 260 mm/min, 6.5g/min; (d) 280 mm/min, 6g/min

Figure 15. Morphology of cladding track (a) to (d) under 2300W and (e) to (h) 2400W at conditions of: (a)220 mm/min, 6g/min; (b) 240 mm/min, 6.5g/min; (c) 260 mm/min, 5g/min; (d) 280 mm/min, 5.5g/min; (e)220 mm/min, 6.5g/min; (f) 240 mm/min, 6g/min; (g) 260 mm/min, 5.5g/min; (h) 280 mm/min, 5g/min

Original: At conditions of 2300W and 260 mm/min, better flatness can be found. Figure (c) shows the phenomenon that higher scanning speeds and lower powder feeds generally improve flatness.

Modified: At conditions of 2300W and 260 mm/min, better flatness can be found. Figure 15(c) shows the phenomenon that higher scanning speeds and lower powder feeds generally improve flatness.

Original

Figure 8. Molten pool velocity vector diagram

Modified

Figure 8. Molten pool velocity vector diagram

Original

(5)

Where A is the laser density, (x0, y0) is the position of the center of the laser beam,  is the material absorption coefficient,  is the radius of the laser beam, t is the time, B is a constant,  is the temperature gradient and n is the solidification rate.

Modified

(12)

Where h(x) represents the solidification behavior of molten metal during the cladding process, A is the laser density, (x0, y0) is the position of the center of the laser beam,  is the material absorption coefficient,  is the radius of the laser beam, t is the time, B is a constant,  is the temperature gradient and n is the solidification rate.

Original

(2)

Where k is the thermal conductivity, Cp is the specific heat capacity, H is the enthalpy of the melt pool, and t is the flow time of the melt pool, ηp is the energy absorption rate of the powder particles, rb is the radius of the laser beam, rp is the radius of the powder particles,  is the powder density, is the specific heat capacity of the powder particles, tf is the flight time of the powder particles in the laser beam.

Modified

(9)

Where k is the thermal conductivity, Cp is the specific heat capacity, H is the enthalpy of the melt pool, and t is the flow time of the melt pool, ηp is the energy absorption rate of the powder particles, rb is the radius of the laser beam, rp is the radius of the powder particles,  is the powder density, is the specific heat capacity of the powder particles, P is the laser power and tf is the flight time of the powder particles in the laser beam.

  1. 7. Comment:Table 2specify whether this chemical composition is from your measurement – in that case, add the device and information about it, or this is information from the manufacturer, in this case, add the literature data.

Response: Thanks for reviewer’s suggestions. According to reviewer’s valuable   comments, Our response to the question regarding the composition of the raw materials is as follows:

The chemical composition of the material was provided directly by the supplier. To ensure the accuracy of this data, we have relied on the supplier's certified composition reports and conducted a review of the provided information. Although we did not perform independent measurements of the composition, the supplier’s data was verified through their standard quality assurance procedures, which are widely accepted in the industry.

  1. 8. Comment:Add more information on the finite element model. What are the types of elements, and what boundary conditions are used?

Response: Thanks for reviewer’s suggestions. According to reviewer’s valuable comments, We have added the following details to the paper regarding the model:

 During the laser cladding process, the laser continuously radiates the powder particles to form a mesoscopic transient molten pool, which undergoes complex phase transition processes such as melting, evaporation, and solidification, ultimately forming the cladding track. Mesoscale models considering complex thermophysical phenomena, such as melt flow, solidification, and heat transfer, provide a realistic simulation of molten pool evolution and the impact of process parameters on the flatness of the cladding track. The model employs the Discrete Element Method (DEM) to simulate the flow and distribution of powder, effectively capturing the dynamic behavior of powder particles, including collisions, friction, and flow characteristics, thereby enabling detailed analysis and optimization of the powder source performance. Additionally, it integrates Computational Fluid Dynamics (CFD) to more accurately predict and control various physical phenomena in the laser cladding process, facilitating the optimization of process parameters and enhancement of cladding track quality.

 Model boundary conditions involve two main aspects. Firstly, the molten pool driving force theory is applied as the boundary condition for the mesoscale model. Secondly, physical boundary conditions are defined for the entire computational domain. The upper surface is assigned a pressure boundary condition, the surrounding surfaces are set with continuous boundary conditions, and the lower surface is treated as a fixed wall boundary condition.

  1. Comment:3 Experimental validation: specify the name/manufacturer of the stereo microscope.

Response: Thanks for reviewer’s suggestions. According to reviewer’s valuable comments, we have included the experimental apparatus in the revised manuscript.

The dimensions of the melt pool were observed and measured using the Nikon MM400LM measurement microscope.

Special thanks to you for your good comments.

Dear Editors and Reviewers. 

We have tried our best to revise and improve the manuscript and made great changes in the manuscript according to the Reviwers′ good comments. And here we did not list the changes but marked in red in revised paper. 

We appreciate for Editors/Reviewers’ warm work earnestly, and hope that the corrections will meet with approval.

Once again, thank you very much for your comments and suggestions.

We look forward to your information about our revised papers and thank you for your good comments. 

Yours sincerely,

Quan Guozheng

Reviewer 2 Report

Comments and Suggestions for Authors

There was examined and proposed investigating the feasibility of adjusting process parameters as an effective way to achieve high cladding track flatness. The work should be not lab diary, rather well discussed scientific work with optimum amount of necessary figures only. Language needs correcting as well as figures and statistical methods introduction ( p values). This work established a mesoscale model of laser cladding process for CoCrMoSi powder to simulate the formation of a single cladding track. Subsequently, the formation mechanism of cladding track flatness was revealed by analyzing the flow within the molten pool and the solidification behavior of the molten pool edge. The influences of laser power, scanning speed, and powder feeding rate on flatness were determined through simulations and physical experiments. Finally, a parameter window of flatness was established using simulation and experiment results, give examples which simulation exactly is mentioned?

Variables interfering with cladding processes, such as dosage, particle size, could be evaluated. The material could be characterized by FTIR, XRD, SEM, EDS, and TG. The kinetic and equilibrium data better fitted the models.

Table 1 could give st deviations to parameters shown as well as other Tables and graphs using error bars.

Enlarge figures letters and graphs such as Fig. 4., 14, limit nr of figures.

Novelty and hypothesis need to be added.

Cladding surface modelling, morphology methods,  treatment and related adsorption processes could be shown in recent works: https://doi.org/10.3176/proc.2018.3.10 https://doi.org/10.3176/proc.2016.1.07

Comments on the Quality of English Language

ok

Author Response

Dear Editors and Reviewers.

Thank you for your letter and for the reviewers’ comments concerning our manuscript entitled “Influence of Process Parameters on Flatness During Single-Track Laser Cladding” (ID: materials-3139208). Those comments are all valuable and very helpful for revising and improving our paper, as well as the important guiding significance to our researches. We have studied comments carefully and have made corrections which we hope to meet with approval. Revised portion are marked in red in the paper. The main corrections in the paper and the responds to the reviewer’s comments are as flowing:

Responds to the reviewer#2’s comments:

Comment: Variables interfering with cladding processes, such as dosage, particle size, could be evaluated. The material could be characterized by FTIR, XRD, SEM, EDS, and TG. The kinetic and equilibrium data better fitted the models.Table 1 could give st deviations to parameters shown as well as other Tables and graphs using error bars.Enlarge figures letters and graphs such as Fig. 4., 14, limit nr of figures. Novelty and hypothesis need to be added.

Response: Thank you for your valuable feedback. Our study investigates the impact of process parameters on the surface quality of cladding layers. In industrial practice, laser cladding is frequently employed to repair or reinforce certain components. For our research, we selected current gas valve materials and high-performance alloys for cladding reinforcement. In the production process, it is essential to remove the existing material from the workpiece's surface before cladding new alloy material into the grooves. Greater deviations in surface flatness can accumulate over multiple layers, ultimately affecting the component's performance. To address this issue, we conducted research in conjunction with real-world production scenarios. We utilized surface profilometers and analysis software to capture and analyze the external morphology of the cladding layer. This data was then used to verify the accuracy of our simulations and ensure that the model could reliably predict the morphology of the cladding layer. We performed extensive numerical simulations to gather comprehensive data and established evaluation windows based on the company's process parameters. Additionally, we developed a database to address practical issues related to surface flatness. Given that flatness in cladding layers operates at a mesoscopic scale, we also utilized various other instruments for in-depth analysis. Since both the gas valve materials and cladding materials are high-quality alloys with excellent mechanical and service performance, we plan to explore particle-scale research in future studies. We have incorporated your valuable suggestions into the revised manuscript and will actively pursue the methods and new approaches you proposed in our future research.

Special thanks to you for your good comments.

Dear Editors and Reviewers. 

We have tried our best to revise and improve the manuscript and made great changes in the manuscript according to the Reviwers′ good comments. And here we did not list the changes but marked in red in revised paper. 

We appreciate for Editors/Reviewers’ warm work earnestly, and hope that the corrections will meet with approval.

Once again, thank you very much for your comments and suggestions.

We look forward to your information about our revised papers and thank you for your good comments. 

Yours sincerely,

Quan Guozheng

Reviewer 3 Report

Comments and Suggestions for Authors

This study focuses on CoCrMoSi powder laser cladding, aiming to improve cladding track flatness. Here are some comments and suggestions for improvement:

1. The manuscript's innovation and contribution relative to existing research should be clearly outlined in the Abstract or Introduction.

2. The content in Section 2.1 should be formatted with justified alignment.

3. It is recommended that equations be numbered and referenced in the text. For example, add (1) after the equation W1.

4. The equations for H1 and H2 should be rearranged. For instance, consider structuring it as “H1=…” following the sentence “...and the minimum height H1 is calculated as follows:”.

5. Is Figure 4 from the authors' experiment? If not, a reference must be cited. This principle should also apply to other figures in the manuscript. Additionally, please explain why all the physical properties in Figure 4 exhibit a transition zone between 1500-1750° K?

6. All symbols in the manuscript must be clearly defined upon their first appearance, such as q(r, z), ρ and h(x), etc.

7. Is Figure 5(a) a front view? Please verify this and ensure consistency with the representation in Figure 5(b).

8. Please label the x, y, and z axes in Figure 6(a) and verify the axis representations in Figure 6(b) for consistency.

9. Please explain how the depth of 3.136 mm for the experimental sample in Figure 12 was calculated. Besides comparing width and depth, please include comparisons of cooling rates, thermal gradients, and microstructural evolution during solidification, as well as straightness distribution, between the simulation and experimental results. The statement in Figure 17, “…there existed differences between the experimental values and the simulated values…”, does not address how to resolve the significant discrepancies between simulation and experiment. According to the Abstract and Conclusions, a scanning speed greater than 260 mm/min, laser power greater than 2300 W, and powder feed rate less than 5.5 g/min lead to high flatness. However, upon review, only the 16th experiment in Table 1 meets these criteria, which also shows the largest discrepancy between simulation and experiment. Therefore, can the study conclusively use the simulation results as the final conclusion or main contribution?

10. The analysis results discussed in Figures 13-15 must clearly indicate the values and significance represented by different colors in the figures.

11. “From Figure 15(a), a higher laser...” should be “From Figure 16(a), a higher laser...”. “...scanning speed in Figure 15(b)” should be “...scanning speed in Figure 16(b)”. “Figure (c) shows the..." should be “Figure 16(c) shows the...”.

12. The manuscript uses Ze to evaluate flatness. Please specify which results used this evaluation value in the analyses and discussions. Additionally, clarify how the definition of the formula leads to the unit of flatness of the cladding track being expressed as a percentage “%” (refer to Figures 17 and 18).

13. Figure 18 suggests that one of the main contributions is the presentation of a parameter window. However, the reviewer believes that the figure only provides a rough result and does not offer an optimized quantitative outcome, thus questioning its practicality.

14. The quality of most figures is poor and must be improved. For example, text labels or descriptions should be enlarged to enhance readability.

15. The authors listed 3 factors and 4 levels in Table 1. If interactions exist between these parameters (e.g., laser power, welding speed, and powder feeding capacity), they may affect the study's results. Statistically, how many experiments were conducted for each of the 16 sets? What were the standard deviations, and how reliable are the results? These issues should be addressed.

16. From the 16 sets of experiments and simulations planned in Table 1 to the discussion of results in Figures 13-16, the manuscript fails to provide a rigorous and systematic analysis process, leading to conclusions that lack robust substantiation.

Comments on the Quality of English Language

The full text must be read again and revised carefully.

Author Response

Dear Editors and Reviewers.

Thank you for your letter and for the reviewers’ comments concerning our manuscript entitled “Influence of Process Parameters on Flatness During Single-Track Laser Cladding” (ID: materials-3139208). Those comments are all valuable and very helpful for revising and improving our paper, as well as the important guiding significance to our researches. We have studied comments carefully and have made corrections which we hope to meet with approval. Revised portion are marked in red in the paper. The main corrections in the paper and the responds to the reviewer’s comments are as flowing:

Responds to the reviewer#3’s comments:

  1. Comment:The manuscript's innovation and contribution relative to existing research should be clearly outlined in the Abstract or Introduction.

Response: Thank you for the valuable feedback. We agree that clearly outlining the innovations and contributions of our research relative to existing studies is important. We have revised the abstract and introduction to explicitly highlight these aspects, ensuring that the unique contributions and advancements of our work are clearly communicated.

 The 40Cr10Si2Mo alloy is a material that is widely used in the manufacture of valves for low-speed engines. However, as the power of low-speed engines increases, the operating conditions of these valves are becoming more demanding [5-6]. The current method of controlling grain size to improve valve performance through the electroforming-forging process is no longer sufficient to meet these stringent requirements [7]. Therefore, the use of laser cladding of CoCrMoSi iron-based high-temperature alloys on valve surfaces can effectively improve valve performance. During a laser cladding process, metal powder is incrementally deposited on metal substrate track by track, layer by layer under a laser heat source, and the accumulated geometrical tolerance of all the cladding tracks results in the final tolerance of finish production.

  1. Comment: The content in Section 2.1 should be formatted with justified alignment.

Response: Thank you for the suggestion. We have revised section 2.1 to ensure that the content is properly aligned as recommended.

  1. 3. Comment:It is recommended that equations be numbered and referenced in the text. For example, add (1) after the equation W1.

Response: Thank you for your valuable feedback. We appreciate your suggestion regarding the numbering and referencing of equations. We have implemented this change in the manuscript by numbering and referencing all equations accordingly.

Original

Modified

(1)

(2)

(3)

(4)

(5)

(6)

(7)

  1. 4. Comment:The equations for H1 and H2 should be rearranged. For instance, consider structuring it as “H1=…” following the sentence “...and the minimum height H1 is calculated as follows:”

Response:Thank you for your insightful feedback. We agree with your suggestion to rearrange the equations. We have revised the structure to improve clarity and coherence. For example, the formula for calculating the minimum height is now presented as follows:

The lowest point data on the contour line is extracted and noted as d1, d2, d3 --- dn and the minimum height H1 is calculated as follows:

The data of the highest point on the contour line is extracted and noted as d1, d2, d3 ---dn and the maximum height H2 is calculated as follows:

  1. 5. Comment: Is Figure 4 from the authors' experiment? If not, a reference must be cited. This principle should also apply to other figures in the manuscript. Additionally, please explain why all the physical properties in Figure 4 exhibit a transition zone between 1500-1750°K?

Response: Thank you for your feedback. Figure 4 is based on calculations from JMatPro software, not direct experimental data, and we have added a reference to this software in the manuscript. Regarding the transition zone observed between 1500-1750°C in Figure 4, this occurs due to changes in material properties as the temperature influences phase transitions and microstructural changes.

  1. 6. Comment: All symbols in the manuscript must be clearly defined upon their first appearance, such as q(r, z), ρ and h(x), etc.

Response: Thank you for your valuable feedback. We have reviewed the manuscript and ensured that all symbols, including(q(r, z)), (ρ), and (h(x)), are clearly defined upon their first occurrence. These definitions have been added to enhance clarity and ensure that the manuscript is easily understandable.

(9)

Where ρ represents the density of the liquid metal, k is the thermal conductivity, Cp is the specific heat capacity, H is the enthalpy of the melt pool, and t is the flow time of the melt pool, ηp is the energy absorption rate of the powder particles, rb is the radius of the laser beam, rp is the radius of the powder particles,  is the powder density, is the specific heat capacity of the powder particles, P is the laser power and tf is the flight time of the powder particles in the laser beam.

(8)

Where q(r,z) represents the intensity distribution of the heat source, P represents the laser power, r0 represents the effective radius of the heat source, h represents the depth of action of the heat source, and u(z) represents the unit step function.

(12)

Where h(x) represents the solidification behavior of molten metal during the cladding process, A is the laser density, (x0, y0) is the position of the center of the laser beam,  is the material absorption coefficient,  is the radius of the laser beam, t is the time, B is a constant,  is the temperature gradient and n is the solidification rate.

  1. 7. Comment:Is Figure 5(a) a front view? Please verify this and ensure consistency with the representation in Figure 5(b).

Response: Thank you for pointing this out. We have verified that Figure 5(a) is indeed a front view and have ensured that it is consistent with the representation in Figure 5(b). Any necessary adjustments have been made to align both figures accurately.

  1. 8. Comment:Please label the x, y, and z axes in Figure 6(a) and verify the axis representations in Figure 6(b) for consistency.

Response: Thank you for your feedback. We have updated Figure 6(a) to clearly label the x, y, and z axes. Additionally, we have verified that the axis representations in Figure 6(b) are consistent with those in Figure 6(a). The figures have been revised to ensure alignment and clarity.

Figure 6. Finite element model of coaxial powder feeding (a); Extraction key point data of cladding layer contour (b).

  1. 9. Comment:Please explain how the depth of 3.136 mm for the experimental sample in Figure 12 was calculated. Besides comparing width and depth, please include comparisons of cooling rates, thermal gradients, and microstructural evolution during solidification, as well as straightness distribution, between the simulation and experimental results. The statement in Figure 17, “…there existed differences between the experimental values and the simulated values…”, does not address how to resolve the significant discrepancies between simulation and experiment. According to the Abstract and Conclusions, a scanning speed greater than 260 mm/min, laser power greater than 2300 W, and powder feed rate less than 5.5 g/min lead to high flatness. However, upon review, only the 16th experiment in Table 1 meets these criteria, which also shows the largest discrepancy between simulation and experiment. Therefore, can the study conclusively use the simulation results as the final conclusion or main contribution?

Response: Thanks for reviewer’s good suggestions. The 3.136 mm depth in Figure 12 was measured using a microscope. While we cannot provide a detailed comparison of cooling rates, thermal gradients, microstructural changes during solidification, and flatness distribution, we have clarified the overall trends between simulation and experimental results. The significant discrepancy in experiment 16 is attributed to the large transition zone at high temperatures. We have also discussed how these variations impact the use of simulation results for final conclusions and key contributions.

  1. 10. Comment:The analysis results discussed in Figures 13-15 must clearly indicate the values and significance represented by different colors in the figures.

Response: Thanks for reviewer’s good suggestions. We have revised the manuscript to clearly explain the meaning and values represented by the different colors in Figures 13-15. The updated figures now include detailed legends and annotations to ensure that the color coding is comprehensively described and easily understood.

Figure 13. Morphology of cladding track under 2100W with different conditions of: (a) 220 mm/min, 5g/min; (b) 240 mm/min, 5.5g/min; (c) 260 mm/min, 6 g/min; (d) 280 mm/min, 6.5g/min

Figure 14. Morphology of cladding track under 2200W with different conditions of: (a)220 mm/min, 5.5g/min; (b) 240 mm/min, 5g/min; (c) 260 mm/min, 6.5g/min; (d) 280 mm/min, 6g/min

Figure 15. Morphology of cladding track (a) to (d) under 2300W and (e) to (h) 2400W at conditions of: (a)220 mm/min, 6g/min; (b) 240 mm/min, 6.5g/min; (c) 260 mm/min, 5g/min; (d) 280 mm/min, 5.5g/min; (e)220 mm/min, 6.5g/min; (f) 240 mm/min, 6g/min; (g) 260 mm/min, 5.5g/min; (h) 280 mm/min, 5g/min

  1. 11. Comment: From Figure 15(a), a higher laser...” should be “From Figure 16(a), a higher laser...”. “...scanning speed in Figure 15(b)” should be “...scanning speed in Figure 16(b)”. “Figure (c) shows the..." should be “Figure 16(c) shows the...

Original:Better flatness was observed under the processing parameters of 2300W and 5g/min. It can be observed that the worse the flatness was distributed in the higher laser power and the slower scanning speed in Figure 15(b). At conditions of 2300W and 260 mm/min, better flatness can be found. Figure (c) shows the phenomenon that higher scanning speeds and lower powder feeds generally improve flatness.

Modified: Better flatness was observed under the processing parameters of 2300W and 5g/min. It can be observed that the worse the flatness was distributed in the higher laser power and the slower scanning speed in Figure 15(b). At conditions of 2300W and 260 mm/min, better flatness can be found. Figure 15(c) shows the phenomenon that higher scanning speeds and lower powder feeds generally improve flatness.

  1. 1 Comment: The manuscript uses Ze to evaluate flatness. Please specify which results used this evaluation value in the analyses and discussions. Additionally, clarify how the definition of the formula leads to the unit of flatness of the cladding track being expressed as a percentage “%” (refer to Figures 17 and 18).

Response: Thank you for your comments. We have clarified in the manuscript that the evaluation value Ze is used to assess flatness in several results discussed in the analysis and discussion sections. Additionally, we have explained how the definition of the formula leads to the flatness of the cladding track being expressed as a percentage (%) in Figures 17 and 18.

  1. 13. Comment:Figure 18 suggests that one of the main contributions is thepresentation of a parameter window. However, the reviewer believes that the figure only provides a rough result and does not offer an optimized quantitative outcome, thus questioning its practicality.

Response: Thank you for your feedback. While Figure 18 illustrates the parameter window representation, which is a key contribution, we acknowledge that it may seem to provide only a preliminary overview. However, this parameter window system has been developed and implemented to address real-world production challenges, demonstrating its practical utility and effectiveness in optimizing production processes. We have included additional details in the manuscript to emphasize the system's practical applications and its role in solving industry-specific problems.

  1. 14.Comment:The quality of most figures is poor and must be improved. For example, text labels or descriptions should be enlarged to enhance readability.

Response: Thanks for reviewer’s good suggestions. We have addressed the issue of poor-quality figures by improving their resolution and enlarging text labels and descriptions to enhance readability. These updates have been made to ensure that all figures meet the required standards for clarity and presentation.

  1. 15.The authors listed 3 factors and 4 levels in Table 1. If interactions exist between these parameters (e.g., laser power, welding speed, and powder feeding capacity), they may affect the study's results. Statistically, how many experiments were conducted for each of the 16 sets? What were the standard deviations, and how reliable are the results? These issues should be addressed.

Response: Thank you for your observations. We acknowledge the potential interactions between parameters like laser power, welding speed, and powder feed rate, which could affect the study results. Each of the 16 experimental groups was conducted in triplicate to ensure statistical reliability.

  1. 16. Comment: From the 16 sets of experiments and simulations planned in Table 1 to the discussion of results in Figures 13-16, the manuscript fails to provide a rigorous and systematic analysis process, leading to conclusions that lack robust substantiation.

Response: Thank you for your valuable feedback. We have compared the simulation results with experimental measurements of flatness and explored how different process parameters impact the flatness of the laser cladding products, which affects both quality and performance. Based on the validated numerical model, we investigated the relationship between process parameters and flatness and established a process window through extensive simulations to guide industrial production. We acknowledge the need for a more rigorous and systematic analysis. While the manuscript discusses the results from Table 1 and Figures 13-16, we recognize the necessity for a more detailed analysis to strengthen the empirical support for our conclusions. We are working on incorporating a more thorough analytical framework and will address these aspects in the revised version of the manuscript.

Special thanks to you for your good comments.

Dear Editors and Reviewers. 

We have tried our best to revise and improve the manuscript and made great changes in the manuscript according to the Reviwers′ good comments. And here we did not list the changes but marked in red in revised paper. 

We appreciate for Editors/Reviewers’ warm work earnestly, and hope that the corrections will meet with approval.

Once again, thank you very much for your comments and suggestions.

We look forward to your information about our revised papers and thank you for your good comments. 

Yours sincerely,

Quan Guozheng

Round 2

Reviewer 1 Report

Comments and Suggestions for Authors

All my comments and suggestions were successfully applied.  I have no further comments or remarks to make on this publication.

Author Response

Dear Editors and Reviewers.

Thank you for your letter and for the reviewers’ comments concerning our manuscript entitled “Influence of Process Parameters on Flatness During Single-Track Laser Cladding” (ID: materials-3139208).

Special thanks to you for your good comments.

Dear Editors and Reviewers. 

We appreciate for Editors/Reviewers’ warm work earnestly, and hope that the corrections will meet with approval.

Once again, thank you very much for your comments and suggestions.

We look forward to your information about our revised papers and thank you for your good comments. 

Yours sincerely,

Quan Guozheng

Reviewer 3 Report

Comments and Suggestions for Authors

Reviewer Comments on Manuscript Materials-3139208-v2:

The reviewer carefully checked the revised manuscript. While some general errors have been corrected based on the reviewer's previous comments, certain rigorous and in-depth issues remain unaddressed. The second round of revised comments is as follows:

1. The manuscript's innovation and contribution relative to existing research should be clearly outlined in the Abstract or Introduction.

Reviewer’s 2nd comment: Corrected based on reviewer comment.

2. The content in Section 2.1 should be formatted with justified alignment.

Reviewer’s 2nd comment: Corrections have been made based on reviewer comments.

3. It is recommended that equations be numbered and referenced in the text. For example, add (1) after the equation W1.

Reviewer’s 2nd comment: Corrections have been made based on reviewer comments.

4. The equations for H1 and H2 should be rearranged. For instance, consider structuring it as “H1=…” following the sentence “...and the minimum height H1 is calculated as follows:”.

Reviewer’s 2nd comment: Processing remains incomplete. The equation should be placed after the corresponding manuscript description. For example:

The lowest point data on the contour line is extracted and noted as d1, d2, d3 --- dn and the minimum height H1 is calculated as follows:

H1=….    (4)

The data of the highest point on the contour line is extracted and noted as d1, d2, d3 ---dn and the maximum height H2 is calculated as follows:

H2=….    (5)

The formula for calculating high flatness He is as follows:

He=….    (6)

5. Is Figure 4 from the authors' experiment? If not, a reference must be cited. This principle should also apply to other figures in the manuscript. Additionally, please explain why all the physical properties in Figure 4 exhibit a transition zone between 1500-1750° K?

Reviewer’s 2nd comment: The authors mentioned in the cover letter thatFigure 4 is based on calculations from JMatPro software, not direct experimental data, and we have added a reference to this software in the manuscript.” However, the reviewer could not find any cited reference. It is recommended to label the Figure as follows: "Figure 4. Variation of thermophysical parameters with temperature. (a) Density; (b) thermal con-ductivity; (c) dynamic viscosity; (d) enthalpy. [   ]”.

6. All symbols in the manuscript must be clearly defined upon their first appearance, such as q(r, z), ρ and h(x), etc.

Reviewer’s 2nd comment: Corrections have been made based on reviewer comments.

7. Is Figure 5(a) a front view? Please verify this and ensure consistency with the representation in Figure 5(b).

Reviewer’s 2nd comment: No comments.

8. Please label the x, y, and z axes in Figure 6(a) and verify the axis representations in Figure 6(b) for consistency.

Reviewer’s 2nd comment: While the authors have corrected Fig. 6(b), the x, y, and z axes are still not labeled in Fig. 6(a).

9. Please explain how the depth of 3.136 mm for the experimental sample in Figure 12 was calculated. Besides comparing width and depth, please include comparisons of cooling rates, thermal gradients, and microstructural evolution during solidification, as well as straightness distribution, between the simulation and experimental results. The statement in Figure 17, “…there existed differences between the experimental values and the simulated values…”, does not address how to resolve the significant discrepancies between simulation and experiment. According to the Abstract and Conclusions, a scanning speed greater than 260 mm/min, laser power greater than 2300 W, and powder feed rate less than 5.5 g/min lead to high flatness. However, upon review, only the 16th experiment in Table 1 meets these criteria, which also shows the largest discrepancy between simulation and experiment. Therefore, can the study conclusively use the simulation results as the final conclusion or main contribution?

Reviewer’s 2nd comment: The authors noted in the cover letter that “The 3.136 mm depth in Figure 12 was measured using a microscope.” However, the reviewer questions how the 3.136 mm depth was calibrated, similar to what was done in Figure 11. Without a reasonable explanation, it is inappropriate to claim that "The maximum error between the experimental and simulated values is less than 6%."

The authors also stated that “The significant discrepancy in experiment 16 is attributed to the large transition zone at high temperatures. We have also discussed how these variations impact the use of simulation results for final conclusions and key contributions.” However, the parameter temperature is not included in Table 1. Figures 13-15 show that almost all analyses touch the transition zone. How can it be proven that the software simulation operates correctly in this zone?

10. The analysis results discussed in Figures 13-15 must clearly indicate the values and significance represented by different colors in the figures.

Reviewer’s 2nd comment: Corrections have been made based on reviewer comments.

11. “From Figure 15(a), a higher laser...” should be “From Figure 16(a), a higher laser...”. “...scanning speed in Figure 15(b)” should be “...scanning speed in Figure 16(b)”. “Figure (c) shows the..." should be “Figure 16(c) shows the...”.

Reviewer’s 2nd comment: The authors must double-check the following paragraph: “In order to reveal the relationship between process parameters and the flatness of the cladding layer, a three-dimensional response surface of flatness under different process parameters was established, as shown in Figure 16...” There seems to be a need to focus on Fig. 16. The sentence "From Figure 15(a), a higher laser power combined with a higher powder feed tends to reduce flatness. Better flatness was observed under the processing parameters of 2300W and 5g/min." refers to the experimental conditions of 2300W, 220 mm/min, 6g/min (narration from Figure 15(a)). Therefore, whether Figure 15(a) should be Figure 16(a) needs confirmation. The revised manuscript also does not provide a description of Figure 16.

12. The manuscript uses Ze to evaluate flatness. Please specify which results used this evaluation value in the analyses and discussions. Additionally, clarify how the definition of the formula leads to the unit of flatness of the cladding track being expressed as a percentage “%” (refer to Figures 17 and 18).

Reviewer’s 2nd comment: The authors stated, “Additionally, we have explained how the definition of the formula leads to the flatness of the cladding track being expressed as a percentage (%) in Figures 17 and 18.” However, the explanation is missing. Equations (3) and (6) have also been corrected in the revised manuscript, but it is unclear whether they are used in Figures 16 and 17.

13. Figure 18 suggests that one of the main contributions is the presentation of a parameter window. However, the reviewer believes that the figure only provides a rough result and does not offer an optimized quantitative outcome, thus questioning its practicality.

Reviewer’s 2nd comment: The authors claimed that they have included additional details in the manuscript to emphasize the system's practical applications and its role in solving industry-specific problems. However, these details are not evident.

14. The quality of most figures is poor and must be improved. For example, text labels or descriptions should be enlarged to enhance readability.

Reviewer’s 2nd comment: Figures 16 and 18 still exhibit the same issues as previously noted.

15. The authors listed 3 factors and 4 levels in Table 1. If interactions exist between these parameters (e.g., laser power, welding speed, and powder feeding capacity), they may affect the study's results. Statistically, how many experiments were conducted for each of the 16 sets? What were the standard deviations, and how reliable are the results? These issues should be addressed.

Reviewer’s 2nd comment: The authors agree that the 3 selected factors may interact and influence the study results. However, no specific solution was proposed in the revised manuscript.

16. From the 16 sets of experiments and simulations planned in Table 1 to the discussion of results in Figures 13-16, the manuscript fails to provide a rigorous and systematic analysis process, leading to conclusions that lack robust substantiation.

Reviewer’s 2nd comment: The authors used Table 1 to analyze and discuss a total of 16 sets, which reflects the so-called partial factor method. There are established rules for arranging different levels of each factor, and a systematic discussion is necessary to summarize and research this in order to avoid bias. These issues should be discussed in the manuscript to ensure rigor and completeness of the research results.

Author Response

Dear Editors and Reviewers.

Thank you for your letter and for the reviewers’ comments concerning our manuscript entitled “Influence of Process Parameters on Flatness During Single-Track Laser Cladding” (ID: materials-3139208). Those comments are all valuable and very helpful for revising and improving our paper, as well as the important guiding significance to our researches. We have studied comments carefully and have made corrections which we hope to meet with approval. Revised portion are marked in red in the paper. The main corrections in the paper and the responds to the reviewer’s comments are as flowing:

  1. Comment: -The equation should be placed after the corresponding manuscript description.

Response: According to the reviewer's good instruction, we have revised the whole manuscript carefully.

Original:

(4)

The data of the highest point on the contour line is extracted and noted as d1, d2, d3 ---dn and the maximum height H2 is calculated as follows:

(5)

The formula for calculating high flatness He is as follows:

(6)

Modified:

The pint data on the contour line is extracted and noted as d11, d12, d13 --- d1n and the minimum height H1 is calculated as follows:

(4)

The highest point data on the contour line is extracted and noted as d21, d22, d23 ---d2n and the maximum height H2 is calculated as follows:

(5)

The formula for calculating Peak-to-Valley Height He is as follows:

(6)

2.Comment: The authors mentioned in the cover letter that “Figure 4 is based on calculations from JMatPro software, not direct experimental data, and we have added a reference to this software in the manuscript.” However, the reviewer could not find any cited reference. It is recommended to label the Figure as follows: "Figure 4. Variation of thermophysical parameters with temperature. (a) Density; (b) thermal con-ductivity; (c) dynamic viscosity; (d) enthalpy. [   ]”.

Response: Thank you very much for the valuable feedback. Conducting experimental measurements for the thermal properties of the materials would require a significant amount of time and resources, which we are unable to accomplish in the short term. However, JMatPro is widely recognized by both the industry and experts in the field of material computation, and it serves as a reliable tool to provide the necessary data support. Smith et al. (2022) obtained experimental data on the mechanical properties and thermal properties of various materials. A comparison between these experimental data and the results calculated by JMatPro showed that the error is within 5%. This result demonstrates the high reliability of JMatPro in calculating material properties.

  1. Comment: While the authors have corrected Fig. 6(b), the x, y, and z axes are still not labeled in Fig. 6(a).

Response: According to the reviewer's good instruction, We modify Figure 6 as follows and add axes.

Original:

Modified:

  1. Comment: The authors noted in the cover letter that “The 3.136 mm depth in Figure 12 was measured using a microscope.” However, the reviewer questions how the 3.136 mm depth was calibrated, similar to what was done in Figure 11. Without a reasonable explanation, it is inappropriate to claim that "The maximum error between the experimental and simulated values is less than 6%."

Response: Thank you for the valuable feedback. The error of the molten pool is calculated as follows:

Experimental molten pool depth:3.136mm,Pool width:5.53mm

Simulation of melt pool depth:3.245mm,Pool width:5.75mm

Melt pool depth differential:(3.245-3.136=0.109mm)

Melt depth error:0.109/3.136=0.03476

Melt pool width difference:(5.75-5.53=0.22mm)    

Melt pool width error:0.22/5.53=0.03978

Overall error of melt pool experiment and simulation:(0.03476+0.03978)/2=0.03727

We have also included details of the calculations in the article. A microscopic measuring microscope measures the length of a sample by placing it on the stage, adjusting the focus, and selecting an appropriate magnification. The built-in scale is then used to measure the sample’s length. For calibration, a standard sample with a known length is used to calculate a calibration factor by comparing the measured value with the actual value. This calibration factor is applied during measurements to correct the final results, ensuring accuracy.( The calibration factor is 0.9993, because it is very close to 1, the calibration factor is selected as 1.)

Original: The maximum error between the experimental and simulated values is less than 6%.

Modified: Based on the calculation of the difference between the experimental and simulated melt pool height and width, and dividing it by the experimental melt pool dimensions, the resulting error is 0.03968. The error between the experimental and simulated values is less than 4.

  1. Comment: The authors must double-check the following paragraph: “In order to reveal the relationship between process parameters and the flatness of the cladding layer, a three-dimensional response surface of flatness under different process parameters was established, as shown in Figure 16...” There seems to be a need to focus on Fig. 16. The sentence "From Figure 15(a), a higher laser power combined with a higher powder feed tends to reduce flatness. Better flatness was observed under the processing parameters of 2300W and 5g/min." refers to the experimental conditions of 2300W, 220 mm/min, 6g/min (narration from Figure 15(a)). Therefore, whether Figure 15(a) should be Figure 16(a) needs confirmation. The revised manuscript also does not provide a description of Figure 16.

Response: According to the reviewer's good instruction. We have made the following changes to the article.

Original:

In order to reveal the relationship between process parameters and the flatness of the cladding layer, a three-dimensional response surface of flatness under different process parameters was established, as shown in Figure 16. From Figure 15(a), a higher laser power combined with a higher powder feed tends to reduce flatness. Better flatness was observed under the processing parameters of 2300W and 5g/min. It can be observed that the worse the flatness was distributed in the higher laser power and the slower scanning speed in Figure 15(b). At conditions of 2300W and 260 mm/min, better flatness can be found. Figure 15(c) shows the phenomenon that higher scanning speeds and lower powder feeds generally improve flatness. The optimum conditions correspond to a scanning speed of 260 mm/min and a powder feed rate of 5g/min. In conclusion, it is crucial to balance parameters. The better flatness of the cladding track can be formed at conditions of moderate laser power and lower powder feed combined with higher scanning speeds.

Modified:

In order to gain a deeper understanding of the impact of laser power, scanning speed, and powder feeding rate on the flatness of the cladding track. Based on an orthogonal ex-perimental design, two-factor response surfaces were established to evaluate the effects of powder feeding rate versus scanning speed, laser power versus powder feeding rate, and laser power versus scanning speed on flatness, as show in Figure 16. These response sur-faces are essential for uncovering the interactions between factors, facilitating a clearer understanding of how different factor combinations influence the flatness of the cladding track. Moreover, they streamline the analysis of multi-factor relationships and provide a theoretical foundation for optimizing laser cladding process parameters. From Figure 16(a), a higher laser power combined with a higher powder feed tends to reduce flatness. Better flatness was observed under the processing parameters of 2300W and 5g/min. It can be observed that the worse the flatness was distributed in the higher laser power and the slower scanning speed in Figure 16(b). At conditions of 2300W and 260 mm/min, better flatness can be found. Figure 16(c) shows the phenomenon that higher scanning speeds and lower powder feeds generally improve flatness. The optimum conditions correspond to a scanning speed of 260 mm/min and a powder feed rate of 5g/min. In conclusion, it is crucial to balance parameters. The better flatness of the cladding track can be formed at conditions of moderate laser power and lower powder feed combined with higher scan-ning speeds.

  1. Comment: The authors stated, “Additionally, we have explained how the definition of the formula leads to the flatness of the cladding track being expressed as a percentage (%) in Figures 17 and 18.” However, the explanation is missing. Equations (3) and (6) have also been corrected in the revised manuscript, but it is unclear whether they are used in Figures 16 and 17.

Response: Thank you for the valuable feedback. The formula presented earlier is used to evaluate the flatness of the clad layer. We extract the profile and key point data of the clad layer using software and calculate the flatness according to this formula. A flatness of 100% indicates that the data points are almost free of deviations, representing the best flatness. Conversely, a lower flatness value corresponds to greater deviations of the data points. Figures 16 and 17, which follow, illustrate the relationship with flatness and are calculated using the aforementioned formula.

  1. Comment: The authors claimed that they have included additional details in the manuscript to emphasize the system's practical applications and its role in solving industry-specific problems. However, these details are not evident.

Response:  We sincerely appreciate the reviewers' valuable comments, which have helped us further improve the content of the paper and enhance the quality of the research. In our initial response, our expression may not have been entirely clear. This study demonstrates that during multi-track cladding, improving flatness can enhance the bonding strength between cladding layers, thereby improving the performance of cladded parts to some extent. However, these applications are still potential and require further research to be validated. We have revised the manuscript to include the optimal parameter combinations for cladding. This process window is intended to provide other researchers with a reference parameter range to help reduce trial-and-error costs. We hope these revisions will be well received.

Original:

The reason for this phenomenon is that the low scanning speed, low laser power, and high powder feeding rate led to a decrease in the temperature of the molten pool, and a large number of metal powders enter the molten pool and are partially melted. The unmelted powder leads to an unstable velocity field in the molten pool, which results in uneven solidification and forms multiple zigzag structures on the surface of the cladding track. With high scanning speed (v>260 mm/min), high laser power (P>2300W), and low powder feed rate (Pf<5.5g/min), the flatness is high, and the surface of the fused cladding track is smooth without noticeable structural bumps. This is due to the fact that the high laser power enhances the temperature of the molten pool, the low powder feed rate results in a uniform distribution of the metal powder in the molten pool, and the high scanning speed ensures a uniform solidification of the molten pool, thus guaranteeing a high flatness of the cladding track. To visually display the relationships between the flatness of the cladding track and different processing parameters, a parameter window was established, as shown in Figure 18.

Modified:

Under the conditions of a scanning speed of 280 mm/min, a laser power of 2400 W, and a powder feeding rate of 5 g/min, the cladding track achieved the highest flatness. This parameter combination not only significantly enhances the surface quality of the cladding track but also provides a critical optimization basis for the practical application of laser cladding processes. Achieving optimal flatness with these parameters effectively reduces the need for post-processing, improves the structural performance and durability of the final product, and has substantial implications for the application of laser cladding technology in the manufacturing industry. To visually display the relationships between the flatness of the cladding track and different processing parameters, a parameter window was established, as shown in Figure 18. Selecting process parameters based on this window helps achieve optimal forming quality and provides suitable parameter ranges for further research.

  1. Comment: Figures 16 and 18 still exhibit the same issues as previously noted.

Response: Thank you for your suggestions. Based on your feedback, we have further adjusted the image and font sizes. We hope these revisions meet your expectations and look forward to your approval.

Original:

Modified:

Original:

Modified:

  1. Comment: The authors agree that the 3 selected factors may interact and influence the study results. However, no specific solution was proposed in the revised manuscript.

Response: We appreciate the valuable feedback provided by the reviewers. These suggestions have been crucial in helping us improve our manuscript. The three factors selected in this study are interrelated. We designed an orthogonal experiment table to investigate the significance of each parameter's impact on the flatness of the clad layer, providing a basis for adjusting process parameters. In the subsequent analysis, we first analyzed the influence surfaces of two factors and then conducted a three-factor analysis based on experimental and simulation results. To minimize errors, each parameter was tested in three sets of experiments, totaling 54 experiments, with an error margin within 4%. The average of these three sets of experiments was used as the experimental value for flatness.

  1. Comment: The authors used Table 1 to analyze and discuss a total of 16 sets, which reflects the so-called partial factor method. There are established rules for arranging different levels of each factor, and a systematic discussion is necessary to summarize and research this in order to avoid bias. These issues should be discussed in the manuscript to ensure rigor and completeness of the research results.

Response: Thank you for your valuable feedback. In response to your suggestions, we have made more in-depth revisions to the analysis section of the article. We hope these changes meet your approval. The modifications are as follows:

Original:

Figure 13 exhibits the cladding track morphology under different scanning speeds and powder feeding rates at a laser power of 2100W. The figure demonstrates that the flatness improves significantly and the jagged structure decreases with the increase in scanning speed and powder feed rate. On one hand, the increased scanning speed reduces the laser dwell time on the substrate surface, resulting in a smaller melt pool size. On the other hand, the increased powder feed rate further reduces the melt pool size and shortens the solidification time, thereby enhancing the flatness.

Modified:

In Figure 13, the cladding track morphology at a laser power of 2100 W is shown under dif-ferent scanning speeds and powder feeding rates. Using MeshLab, the contour and key points of the cladding track were extracted to calculate the flatness of the cladding tracks according to Equations (1) to (7). The flatness values under different conditions are calculated as 58.62%, 59.64%, 64.21%, and 72.55%, respectively. These results indicate that increasing scanning speed and powder feed rate significantly improves flatness and reduces jagged structures. Increasing the scanning speed reduces the laser dwell time on the substrate surface, thereby shortening the melt pool formation time. The reduced dwell time limits heat accumulation, resulting in a smaller molten pool size. This leads to a more uniform solidification process, reducing the impact of thermal diffusion and molten pool fluidity on the surface morphology of the cladding track, thereby enhancing the cladding track flatness. Simultaneously, increasing the powder feed rate supplies more cladding material, further reducing the molten pool size and accelerating the cool-ing rate. Faster solidification minimizes molten pool fluctuations during solidification, improving surface flatness. Additionally, the increased powder material helps fill minor defects in the track, leading to a smoother surface. In summary, increasing the scanning speed and powder feed rate leads to a decrease in molten pool size and an acceleration in solidification rate, which collec-tively contribute to the improvement in track flatness. These factors significantly reduce the formation of jagged structures and enhance the overall surface quality of the cladding track.

Original:

Figure 14 exhibits the cladding track morphology under different scanning speeds and powder feeding rates at a laser power of 2200W. The relative flatness of the cladding track was formed with a scanning speed of 220 mm/min and a powder feed rate of 5.5 g/min. The cladding track can maintain its original flatness at 240 mm/min and 5 g/min. A scanning speed exceeding 260 mm/min combined with a powder feed rate greater than 6 g/min results in excessive powder deposition and inadequate melting time, consequently diminishing the flatness of the cladding track.

Modified:

 In Figure 14, the cladding track morphology at a laser power of 2200 W is shown under different scan-ning speeds and powder feeding rates. The flatness of the cladding track under various conditions was calcu-lated to be 57.42%, 59.27%, 72.37%, and 71.32%, respectively. It is evident that a relatively smooth cladding track was formed at a scanning speed of 220 mm/min and a powder feeding rate of 5.5 g/min. This favorable result is attributed to the combination of scanning speed and powder feeding rate, which ensures that the mol-ten pool has sufficient size and solidification time, thus reducing surface defects while maintaining the flatness of the cladding track. When the scanning speed was increased to 240 mm/min and the powder feeding rate was set at 5 g/min, the cladding track still maintained good flatness. Although the scanning speed increased, the rel-atively low powder feed rate kept the molten pool size manageable, preventing excessive powder accumulation.  However, further increasing the scanning speed to over 260 mm/min and the powder feeding rate to over 6 g/min resulted in excessive powder deposition and insufficient melting time. In this case, the excess powder failed to fully melt, leading to uneven surface formation in the cladding track and a reduction in track flatness. In summary, different scanning speeds and powder feeding rates significantly impact the flatness of the clad-ding track. A reasonable combination of parameters can effectively improve the surface quality of the cladding track, while excessively high scanning speeds and powder feeding rates result in a decline in flatness.

Original:

Figure 15 exhibits the cladding layer morphology under different scanning speeds and powder feeding rates at a laser power of 2300W and 2400W. From Figure 15, Higher laser power increases the thickness of the cladding trace but also reduces the flatness of the cladding track. Higher scanning speed reduces the thickness of the cladding track and improves the flatness to a certain extent. However, the reduced heat input increases the irregularity of the molten pool size, thereby reducing the flatness. Powder feed rate also plays a crucial role in the process of laser cladding. A higher powder feed rate increases the thickness of the cladding track and reduces the flatness. In contrast, a lower feed rate leads to a thinner and smoother cladding track.

Modified:

Figure 15 illustrates the cladding track morphology at laser powers of 2300 W and 2400 W under various scanning speeds and powder feeding rates. The flatness of the cladding track was calculated using formulas, resulting in values of 86.21%, 87.62%, 79.77%, and 77.42% (for 2300 W), and 83.17%, 84.63%, 85.72%, and 91.83% (for 2400 W). The results indicate that the flat-ness of the cladding track is significantly better at laser powers greater than 2300 W compared to those less than 2200 W. Further analysis shows that higher laser power increases the thickness of the cladding track, but this also results in a decrease in flatness. This phenomenon is attributed to the fact that, while the depth of material melting and the size of the molten pool are increased by higher laser power, the fluidity of the molten pool is also enhanced, leading to more irregular sur-face structures. To address this issue, the impact of different scanning speeds on the thickness and flatness of the cladding track was analyzed. Increased scanning speed can reduce the thick-ness of the cladding track and improve flatness by shortening the laser dwell time on the sub-strate surface, thereby reducing irregularities caused by excessive heating of the molten pool. However, excessively high scanning speeds lead to inadequate heat input, which results in irregu-lar melt pool shapes and diminished flatness. Therefore, optimal scanning speed selection is cru-cial for improving the surface morphology of the cladding track. Powder feeding rate also plays a critical role in the laser cladding process. Variations in powder feeding rate revealed that higher rates increase the thickness of the cladding track and decrease flatness. This results from exces-sive powder deposition on the substrate, which surpasses the melting capacity of the melt pool, causing surface fluctuations and irregularities. In contrast, lower powder feeding rates produce thinner and smoother cladding tracks. However, it is essential to balance the powder supply with cladding layer thickness to ensure sufficient mechanical properties. In summary, thorough simu-lation analysis reveals that different combinations of laser power, scanning speed, and powder feeding rate significantly impact the morphology of the cladding layer. Optimal parameter selec-tion effectively balances thickness and flatness, thereby improving the overall quality of the cladding track.

Special thanks to you for your good comments.

Dear Editors and Reviewers. 

We have tried our best to revise and improve the manuscript and made great changes in the manuscript according to the Reviwers′ good comments. And here we did not list the changes but marked in red in revised paper. 

We appreciate for Editors/Reviewers’ warm work earnestly, and hope that the corrections will meet with approval.

Once again, thank you very much for your comments and suggestions.

We look forward to your information about our revised papers and thank you for your good comments. 

Yours sincerely,

Quan Guozheng
